# The Overview of the Conservation and Renewal of the Industrial Belgian Heritage as a Vector for Cultural Regeneration

**Jiazhen Zhang** [1], **Jeremy Cenci** [1,*] , **Vincent Becue** [1] **and Sesil Koutra** [1,2]

1   Faculty of Architecture and Urban Planning, University of Mons, Rue d' Havre, 88, 7000 Mons, Belgium; Jiazhen.zhang@umons.ac.be (J.Z.); Vincent.becue@umons.ac.be (V.B.); sesil.koutra@umons.ac.be (S.K.)
2   Faculty of Engineering, Erasmus Mundus Joint Master SMACCs, University of Mons, 7000 Mons, Belgium
*   Correspondence: Jeremy.cenci@umons.ac.be; Tel.: +32-498-79-1173

**Abstract:** Industrial heritage reflects the development track of human production activities and witnessed the rise and fall of industrial civilization. As one of the earliest countries in the world to start the Industrial Revolution, Belgium has a rich industrial history. Over the past years, a set of industrial heritage renewal projects have emerged in Belgium in the process of urban regeneration. In this paper, we introduce the basic contents of the related terms of industrial heritage, examine the overall situation of protection and renewal in Belgium. The industrial heritage in Belgium shows its regional characteristics, each region has its representative industrial heritage types. In the Walloon region, it is the heavy industry. In Flanders, it is the textile industry. In Brussels, it is the service industry. The kinds of industrial heritages in Belgium are coordinate with each other. Industrial heritage tourism is developed, especially on eco-tourism, experience tourism. The industrial heritage in transportation and mining are the representative industrial heritages in Belgium. There are a set of numbers industrial heritages are still in running based on a successful reconstruction into industrial tourism projects. Due to the advanced experience in dealing with industrial heritage, the industrial heritage and the city live together harmoniously.

**Keywords:** conservation; cultural regeneration; industrial heritage; (urban) regeneration; renewal

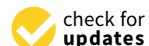

## 1. Introduction

The genesis of the notion of the cultural landscape is more likely laid on the multi-layered dynamic interrelations—both spatially and historically—of human intervention and natural processes to adjust its function to the changing community demands. Its understanding provides a way to bring the tangible and the intangible qualities of a shared environment and to enable its regeneration [1]. Industrialization generated significant changes in the urban and social landscape, including greater densities and the urbanization of the natural and rural environment; population moves and demand for reorganization of modern communities. However, over the past decades' phenomena, such as the globalization, deindustrialization, the urbanization and the economic (re)conversion had profound effects on traditional industrial areas leading to a vast array of obsolete and former industrial facilities generated by them [2].

During last decades, several studies have analyzed and documented the remnants of the industrial society [3,4] and emphasized the necessity of considering post-industrial landscapes in the city planning and the industrial heritage as a resource and an integral part of collective identity, while its preservation as 'vital' and vector for the historical identity.

At the beginning of the 21st century, it has been acknowledged that industrial heritage is understood and interpreted at the level of the landscape and of societies. That broader interpretation of the industrial heritage "focusses on the remains of the industry—sites, structures and infrastructure, machinery and equipment, housing, settlements, landscapes, products, processes, embedded knowledge and skills, documents and records, as well as the use and treatment of this heritage in the present". It should comprise "not only the remains

of the Industrial Revolution, but also the traditional precursors from earlier centuries that reflect increased technical specialization, intensified productive capacity, and distribution and consumption beyond local markets, hallmarks of the rise of industrialization" [5].

The Industrial Revolution represented one of the most significant evolution in the history of mankind [6], important technological advances being registered in that period. In fact, the emergence of the Industrial Revolution is dated dynamically for each country and it is an on-going process in the 20th century i.e., quantitatively approaching; United Kingdom (1750/60), France (1780), Belgium (1790), Germany (1795), United States (1800), Russia (1850), Japan (1860), Brazil (1929), India (1947), China (1953) and so on. (Albrecht 2012). Therefore, each country needs to define and document its sub-frames of the industrialization process, though in the general framework of the global periodization system [5].

In this paper, we introduce and re-define the basic contents of the related terms of industrial heritage and discuss the relationship between industrial heritage and industrial sites. Exploring the particularities of the Belgian industrial heritage, the development, and protection path of the Belgium industry were discussed. Then, we propose a classification of the Belgian Industrial heritage, while at the end, we raise our conclusion for current conservation and renewal progress in Belgium. The paper is structured accordingly to the motivation and significance of industrial heritage; firstly, an overview and the understanding of the main terms and concepts of the work are provided, in the second section, an exhaustive focus of the research of the Belgian case is discussed; the third part of the paper presents historical landmarks and highlights important points related to the protection of the Belgian industrial heritage, from the perspective of classification and region distribution, while the last ones highlight the main findings and discusses the perspectives for future work.

## 2. State-of-the-Art-Analysis of Industrial Site Reconstruction

In this research, the term 'Industrial site' refers to the land for abandoned factories, workshops, handicraft workshops, production sites for construction and installation, and slag discharge sites.

The industry has an important contribution to city development by promoting the expansion of urban scale, which makes the former villages and towns rapidly transform into cities and changed the human lifestyle. Industrial production provides the opportunities for employment, attracts people to the city and promotes the development of other industries, such as infrastructure construction, transportation, service industry, and so on [7,8]. The industrial development and prosperity of the city, especially the regional central city, also includes its basic support for the city's tertiary industry. The expansion of urban industrial facilities has a direct impact on the land use of the primary industry and is closely connected to the change of the overall urban industrial structure.

### 2.1. Industrial Site

There are a series of explanations of 'industrial site' in the mainstream academic literature. In this study, the concept has a twofold understanding: from the one side, it implies the site and the production activities; while at the other it refers to the remains of human activities. From the perspective of history, aesthetics, and anthropology, we observe the outstanding universal value of human engineering or the joint project between man and nature, and archaeological sites. The characteristics of the sites main present an incomplete state. Once the factory stopped manufactured and was abandoned, their facilities in tangible and intangible were deficient more or less. It occupies a certain range of land space at the same time. In this research, the range of industrial sites includes the former industrial land, that is, the land for industrial production, the land for transportation, and storage related to industrial production, for instance, mines, quarries, factories, railway stations, wharves, industrial waste dumps and so on. Industrial sites, especially those

in the center of the city, occupy plenty of land space in the city, cut off the infrastructure network of the city, and hinder the normal development of the city [9].

The traditional manufacturing method was rough and without environmental reflection and urban planification. Along with the economic upgrade and sustainable appearance industrial sites appeared. The main motivations for industrial sites emerging are as follows [10]:

- With the progress of science and technology and productivity, the social and economic structure has undergone profound changes. Traditional products and industries have been eliminated because they cannot meet the market demand, and factories are forced to stop production and close down.
- The depletion of natural resources, which led to the abandonment of mining and related activities. Industrial land became a heritage because of the cessation of production.
- Industrial pollution and consequences to the soil and groundwater in the plant area in the long-term production. After the relocation of these enterprises, the pollutants left in the factory area cannot be treated for a short time, and the land is no longer suitable for use because of the major threat to human health and has become an industrial heritage.

The setting of industrial land directly affects the layout of urban planning. The passenger and freight flow required by industrial production also has a significant impact on the organization of urban road traffic. The adjustment of industrial land will also affect the organization and layout of urban traffic. While the industry actively promotes urban development and social progress, it also produces many negative problems, which affect the further development of the city [11,12].

Industrial land generally includes the following types:

- The production workshop, warehouse, and ancillary facilities of industrial and mining enterprises. It includes dedicated railway, wharf, and road land, excluding open-pit mining land, which is classified into water area and other lands.
- The storage land: warehouse, stockyard, packaging workshop, and its ancillary facilities of storage enterprises.
- The transportation land: railway, highway, pipeline transportation, port, airport, and other urban external transportation and ancillary facilities.
- The land for municipal public and other facilities and infrastructure.

*2.2. The Conservation and Renewal of Industrial Sites*

During third Industrial Revolution, which began at 50s, the western developed industrial countries have undergone profound economic structural changes. Traditional industries began to decline, once prosperous factories gradually lost their vitality, leaving abandoned or idle industrial sites. These industrial sites have become the pronoun of decline, poverty, crime, and pollution, and become the brownfield of the urban landscape. The rejuvenation of industrial sites has become an integral part of urban development and social progress in modern societies.

From the 1970s to the 1980s, the transformation of the material environment was mainly emphasized. In the early stage, it was mainly pushed to reconstruction. In the later stage, it began to pay attention to the protection of industrial remains and the mixed transformation of multiple functions. In the 1990s, after a long period of exploration, western countries gradually formed a comprehensive and integrated transformation method, sustainable development to solve the problem of industrial sites [13].

*2.3. Industrial Heritage*

The term of 'industrial heritage' refers to 'the physical remains of the history of technology during the manufacturing and mining activities, as well as power and transportation infrastructure'. Another definition expands this scope involving social activities related to the industry such as housing, museums, education, or religious worship, highlight the interdisciplinary character of the term [14].

In 2003, the International Committee for the protection of industrial heritage (TICCIH) signed the Nizhny Tagil Charter for the Industrial Heritage in Russia. This document clarified the definition of industrial heritage, by emphasizing its value and importance and became one of the most significant authoritative files in industrial heritage protection. In fact, within this document the 'industrial heritage' is explained as the 'evidence of intensively industrialized activities with profound historical evidences providing a unique sense of identity' ([15,16]).

In 2011, the 17th Congress of the International Council of Monuments and Sites adopted the Dublin principles, a common principle for the protection of sites, structures, areas, and landscapes of industrial heritage sites. The Dublin principles pointed out that the value of industrial heritage lies in the production structure or site itself, including the material composition of machinery and equipment, industrial landscape, literature, as well as the intangible records in memory, art, and customs. The Dublin principles clarified the connection between the cultural and natural environment of industrial heritage reflections. It improved industrial heritage protection to a new level.

In the past 40 years, people's understanding of industrial history and industrial heritage has gradually deepened with the belief that industrial structures, technological processes, and production tools, as well as industrial sites, landscapes, and intangible cultural heritage, are of great value [16], which can be summarized as historical value, scientific and technological value, artistic value, social and cultural value, and economic value.

### 2.4. The Relationship between the Renewal of Industrial Sites and the Conservation of Industrial Heritage

Industrial sites and industrial heritage are closely linked as a whole (Figure 1).

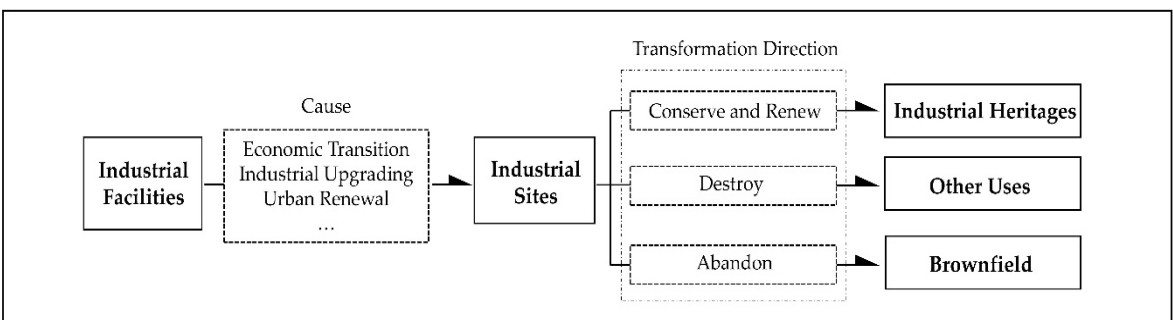

**Figure 1.** The internal relationship between industrial sites and industrial heritage.

This study suggests that the reconstruction of industrial sites is an inevitable trend but also an 'obligation' of social and urban development of modern societies, which requires, nonetheless, a thorough knowledge and political responsibility. Undoubtably, the interventions towards this redevelopment have significant impacts and an important role for the economic regeneration of these derelict areas.

After the 1980s, U.K. and Germany, that are leading industrial countries, have realized from the early demolition and construction that this kind of blind pursuit of material environment transformation is one-sided and its huge damage to the industrial heritage, thus ultimately completing the comprehensive rejuvenation of industrial sites, and the overall solution to the decline of these sites and their effective conservation, they set up strategies of their reconstruction together along with the axis of the principles of sustainable development [17,18].

The precious resources are not only reflected in the material and economic value of industrial heritage itself, but also reflected in the comprehensive historical value, science, technology, society, and art. Due to the differences in scale, use, age, and historical development, industrial heritage presents unique industrial characteristics, which cannot be copied. This kind of uniqueness is extremely rare and scarce in the lack of its characteristics

of the contemporary urban landscape. The transformation of industrial sites should also be understood as the protective transformation of industrial heritage. Through the transformation of industrial sites, we can improve the internal and external space environment of industrial heritage, increase the visibility and social and cultural influence of industrial heritage, increase protection funds, and enhance the public interest and appreciation for its value.

In the understanding of industrial heritage conservation, we should also consider the traditional protection thinking of existing strategies, as it consists of a complex process demanding a comprehensive treatment of the increasingly complex relationship between people and the site [19]. Besides, some heritages are still in industrial production and use, for instance, the crystal discovery Val saint lambert in Belgium, a crystal museum where visitors can not only have a guide includes a demonstration of glass production but also watch glass-blowers working in the heat. The factory is still running and has a significant position in the crystal industry in Belgium. At the subsequent section, a thorough analysis of the highlights of the industrial heritage in Belgium is provided.

### 3. Overview of the Development of the Industrial Heritage in Belgium

After a general overview of the main terms of the industrial heritage and its related issues, the motivation and significance of the conservation and renewal of the industrial heritage have been established. A particular focus is provided for the country, as one of the most important parts of the industrial system in the world, Belgium.

In Continental Europe, Belgium is the first country, which absorbs and imports the U.K.'s Industrial Revolution achievements, in 1802 starting with the textile's industrialization (cotton) in the region of Ghent and the wool in Verviers [20]. Before this period, the country had an important industrialized activity focussed mainly on trading; on the other hand, at the region of Flanders, the textile production had been flourished, while in Walloon region we could observe mainly the coal mining; these two branches have been the keys for industrialization in Belgium for many years.

### 3.1. Overview of the Industrial Development in Belgium

Belgium has a rich industrial past. An important landmark for Belgian heritage to mark out has been the first steal engine, inspired by T. Newcomen, which was succeeded by another one around the regions of Mons and Charleroi. During the period of the French domination (1795 to 1814), coal was mainly mined in the Borinage surroundings (province of Hainaut). From 1795 to 1814, coal was mined in the Borinage (southwest of Mons) to feed Paris via the river network, under the French administration. Based on the foundation of the French Revolution, the Industrial Revolution of Belgium was engine [21]. Around 1800s, William Koclear set up the first factory in Seraing, near Liege, from which the Belgian Industrial Revolution began (Figure 2).

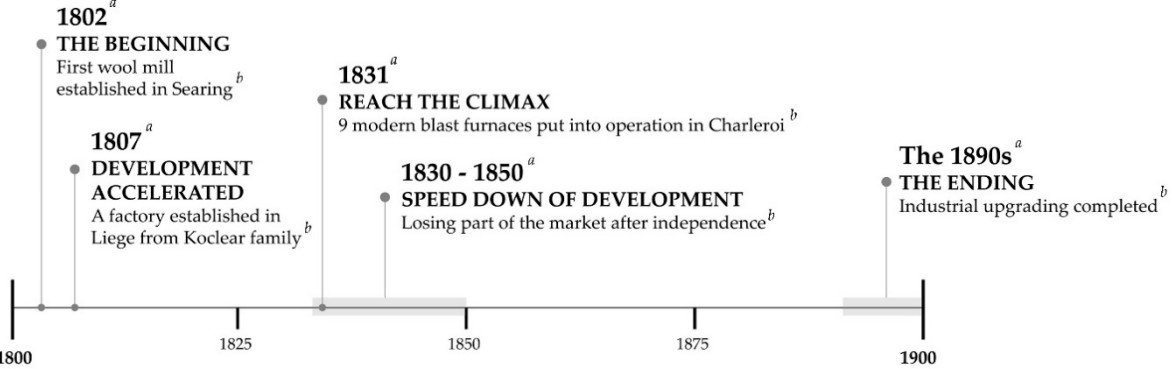

**Figure 2.** Significant landmarks in the history of industrial development in Belgium (a. Date and event derived from Miklas Tich and Roy Porter (1996). b. Stage was defined by the author and references based on historical events).

The regional distribution of the Industrial Revolution center in Belgium is presented in Table 1.

**Table 1.** Belgian Industrial Revolution centres.

| District | City [a] | Field [b] | Representative Industry |
|---|---|---|---|
| Liege | Verviers | Industry promotion | Machinery |
| Charleroi | Mons | Industry promotion | Coal |
| - | Ghent | Industry promotion | Textile |
| - | Antwerp | Service | Port |
| - | Brussels | Service | Capital |

[a]. City and its representative industry derived from Miklas Tich and Roy Porter (1996). [b]. Field was defined by the authors.

But Belgian heavy industry took off during the Dutch rule of the country from 1815 to 1830. Notwithstanding, Belgian heavy industry took off during the Dutch rule of the country of the period 1815 to 1830. At the same period, William I, supported by J. Cockerill, exploited the activities around the textile machinery. At the end of the 18th century. Due to the introduction of the steam engine. Belgium was the first continental country into the Industrial Revolution. It started by industrializing the textile industry in Ghent (cotton) and Verviers (wool).

As the first industrialized country on the European continent, Belgium quickly followed in Britain's footsteps. However, only by the end of the 19th century, did industry and industrialized work become more important than agriculture and artisanal production. This becomes clear when looking at the employment figures. In 1846, the industrial census noted 90,000 industrial workers, or 4% of the total working population, whereas, by 1910, the industry employed nearly half of the working population. This perception of Belgium being a rapidly industrializing country was mainly due to coal mining and the metal and textile industry; the remains of Belgium's industrial past are characterized by a great diversity [22]. Before this period, Belgium was a traditional industrial country in Europe [23].

The first Industrial Revolution witnessed clusters of industrial activities located near the natural resources and raw materials. Liège already had exploited proto-industry with forges and gun manufacturing, while Charleroi was predominated in nail factories, so they were able to embrace coal mining, steel, and other metal activities extensively, which led to population move from the countryside for job opportunities. From the other side, the canal of Brussels-Charleroi was inaugurated in 1832 by linking the mines to the North Sea through the city of Brussels (Figure 3) [24].

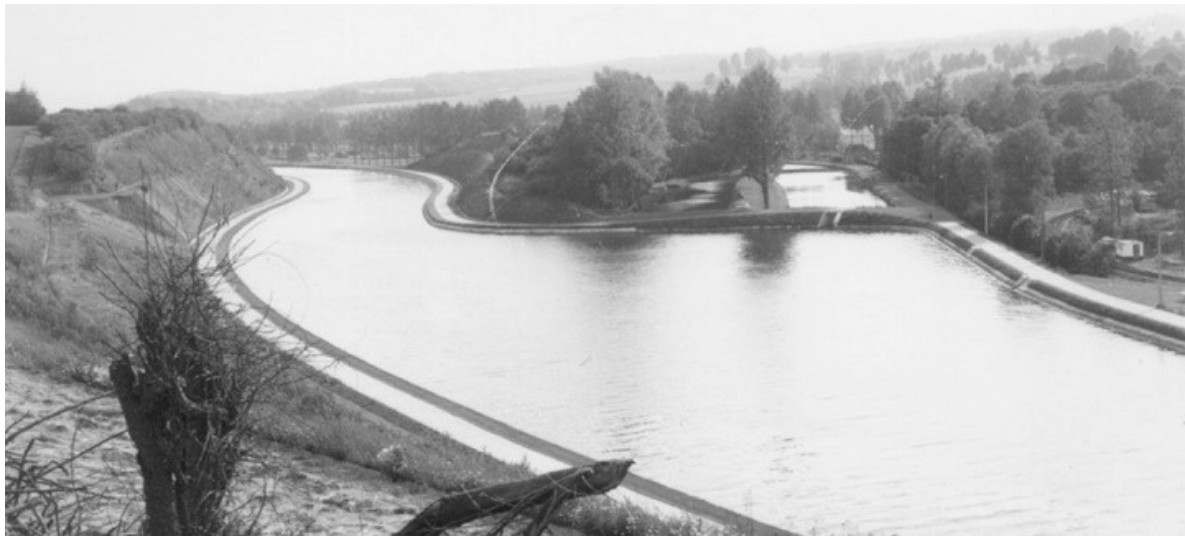

**Figure 3.** The Brussels-Charleroi canal, 1966.

At the same time, the center of Belgium, and in particular the part of its connection with the city of Antwerp, was gaining momentum with new industrialized activity. Antwerp and the port of Liege were gradually becoming the key logistics pivot of Belgium even for the whole of Western Europe (Figure 4) [25].

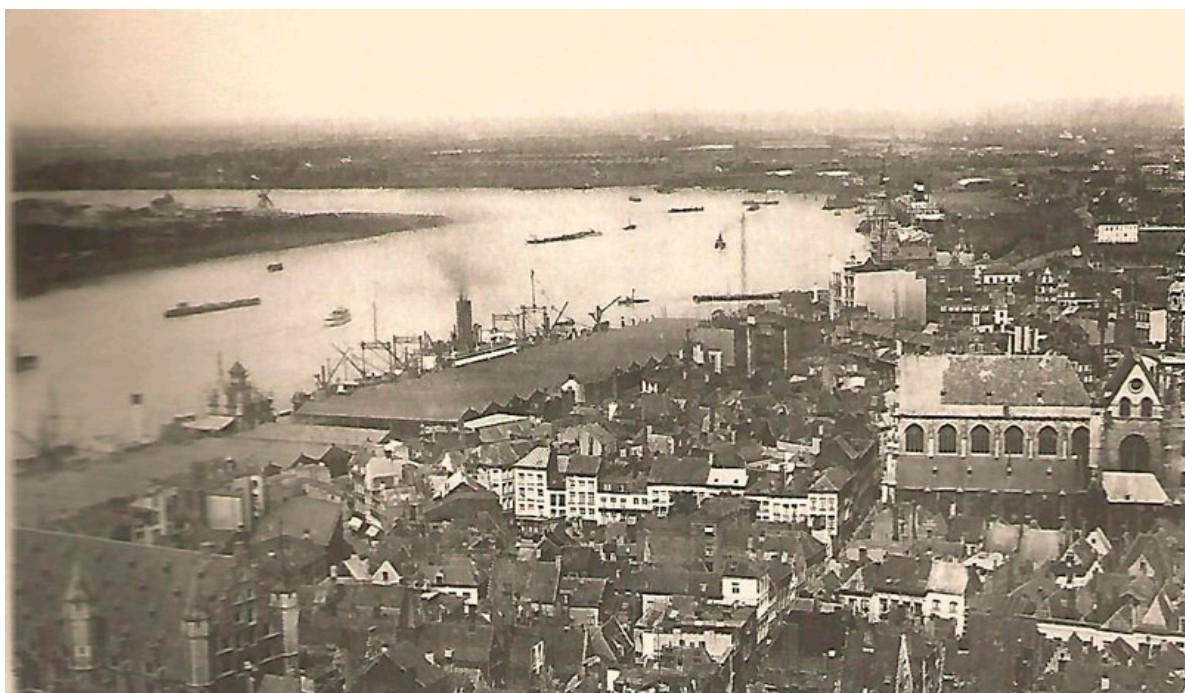

**Figure 4.** Antwerp—an industrious historic city, 1863.

After the First World War, the arrival of Ford and General Motors to Antwerp and Renault to Brussels was an important step for the industrialized activities of Belgium, while in 1917 the coal finding in Limburg (eastern Belgium) marked a new era. However, this evolution was ceased during the internal crisis of 1930s [24].

After the Second World War, the 'heavy' industrialized and intensive activities in Walloon region seemed competitive; nonetheless, very soon the weaknesses and the need for modernization became apparent [24].

*3.2. Inspection and Research Institutions for Industrial Heritage*

The industry was engine by economic power primarily. However, the development path of industrial heritage was influenced by industrial heritage inspection and research institutions [26], for instance, the UNESCO assessed a set of industrial heritage list to protect the industrial sites in the range of the world. The industrial sites on the list can take higher-level protection than before. To have a comprehensive understanding of the development status of industrial heritage in each region, it is also necessary to understand the status quo of their own internal industrial heritage research institutions. The Industrial Revolution of the U.K. spends 50 years to spread into Belgium. Nonetheless, Belgium started an industry heritage investigation, followed by the U.K. tidy [27].

In 1970, Georges van den Abeelen created the industrial archaeology center, it was a trigger for industrial inspection and record subject in Belgium [25]. Meanwhile, it was where the exhibition man and machine were held in Brussels two years later supported by historians and archaeologists from different fields.

In Flanders, the VVIA (1978) had an important role on national level and eventually represented the whole country at the issue of inspection and recording of the Belgian industrial heritage [28]. After an extensive analysis of research projects, and reports in 1984, based on the VVIA, the PIWB and TICCIH Belgium were established [29] as a triangular

relationship. This is a unique situation in Western European countries (Table 2). Each institution is doing inspections and records for industrial heritages in their regionals.

**Table 2.** Schedule of main industrial heritage inspection and research institutions in Western European countries.

| Location | Organization | Abbreviation [a] | Established Time [a] |
|---|---|---|---|
| The U.K. | Association for Industrial Archaeology | AIA | 1973 |
| German | The International Committee for the Conservation of the Industrial Heritage Deutsch | TICCIH Deutsch | 1978 |
| France | Le Comité d'information et de liaison pour l'archéologie, l'étude et la mise en valeur du patrimoine industriel | CILAC | 1979 |
| The Netherlands | Federatie Industrieel Erfgoed Nederland | FIEN | 1984 |

[a]. Abbreviation and Established time derived from its online website, see Appendix A Table A1.

Take the PIWB as an example, it was established at the time of the rise of industrial heritage protection in Belgium. In the early stage, its work mainly focused on the publicity of the restored sites and promoting the social protection of potential sites. With the advancement of research and the development of industrial tourism, the work of the PIWB has become diversified and the related fields have become wider. They began to pay attention to the relationship between industry and human beings. Meanwhile, the concept of industrial archaeology was expanding considerably, calling for vigilance extended to other fields—ancestral technologies and proto-industries, tangible as well as intangible heritage, oral history.

### 3.3. Industrial Heritage in Belgium

As previously mentioned, 'industrialization' has had different impacts leading to significant changes in the urban, social and cultural environment (for instance, greater density and more compact urban areas, population moves, etc.)., thus contributing to the typical 20th-century urban settlement [30]. The classification of a territory as 'industrial' implied a qualitative perception, in which territory and industrial infrastructures were analyzed from a functional, cultural, and historic angle [31]. In this sense, and according to Borsi [32] the industrial landscape is "the landscape resultant from a thoughtful and systematic activity of man in the natural or agricultural landscape with the aim of developing activities related to the industry". This definition enabled the recognition of an entire landscape as a single "element", allowing the expansion of the conception of its conservation to accommodate "recognized patterns of activity in time and place" [33].

Based on the ERIH, this research categorized the industrial heritage of Belgium (Table 3). Listing the most important elements of the Belgian industrial heritage, we observe that Belgium scatters its heritage in every corner with a variety of categories (Table 4).

**Table 3.** Main categories of the Belgian industrial heritage.

| Heritage Category |
|---|
| Industry and War |
| Iron and Steel |
| Landscape |
| Mining |
| Production and Manufacturing |
| Water |
| Communication |
| Textiles |
| Service and Leisure Industry |
| Paper |
| Transport |

**Table 4.** Main elements of the Belgian industrial heritage.

| Industrial Heritage | Description | Illustration |
|---|---|---|
| Blégny Mine | Located between Liège and Maastricht, Blegny-Mine is one of the four authentic coal mines in Europe with underground galleries accessible for the visitors through the original shaft |  |
| Museum Plantin-Moretus | The world's oldest surviving printing works is a work of art in itself |  |
| Val St-Lambert | The Belgian glassware that took the world by storm a century ago |  |
| Speelkaartmuseum | The huge scale of machines once required to make playing cards |  |
| Le Bois du Cazier | A slag heap in archetypal post-industrial Charleroi |  |
| Canals of the Borinage | Remarkable ship-lifts, old and new |  |
| Centre Touristique de la Laine et de la Mode | A fascinating textile museum |  |

Most of them were renewed properly and 6 of them were added to the world heritage (UNESCO) (Figure 5) [34]. On the one hand, redeveloping these heritages to cultural or other type of activities through adaptable architectonic interventions, while on the other hand, surveying and analyzing their conservation and renewal paths are a benefit for urban regeneration and the science of human settlement. UNESCO promotes at the same time the identification, conservation, and preservation of the world heritage as an outstanding

value to humanity. Europe is particularly well represented on the list of heritage sites, as is Belgium (Figure 6) [35].

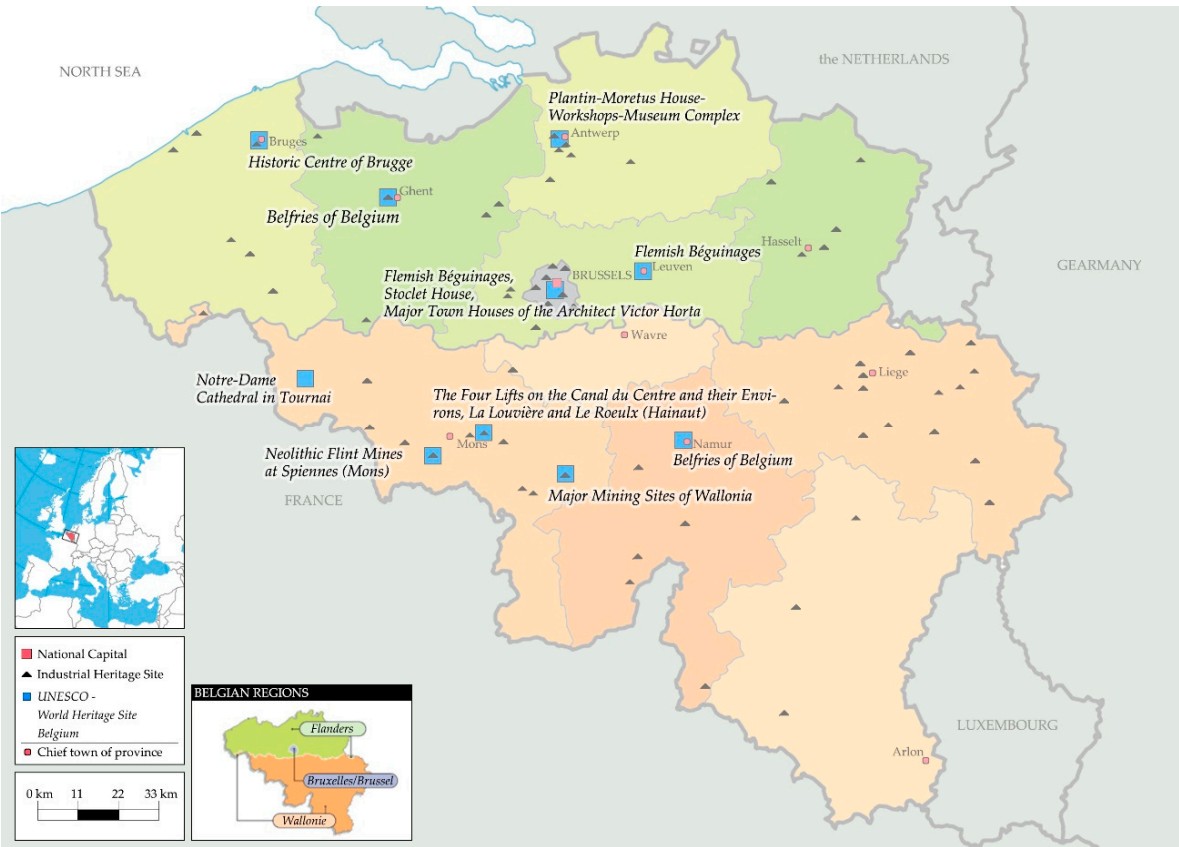

**Figure 5.** The UNESCO-world heritage and Industrial heritage landmarks in Belgium.

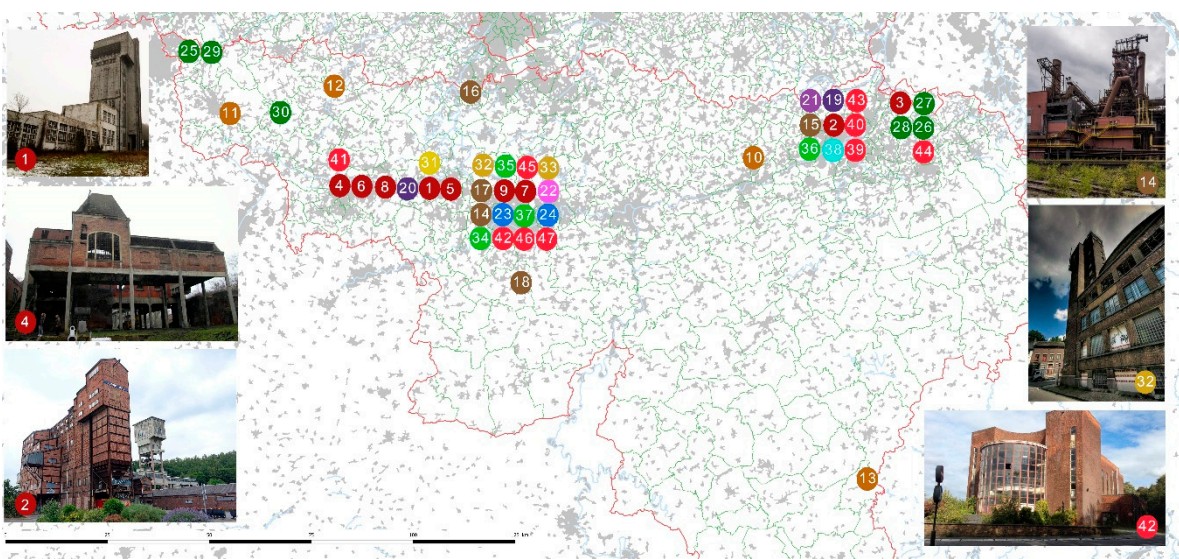

**Figure 6.** Industrial heritage in Wallonia counting.

In addition, we can identify many industrial wastelands in Walloon territory. Yet, there has not been a detailed existing inventory of an architectural and heritage interest but only according to several categories resulting from past production. We can easily see that

the latter are located on the Croissant Houiller with the major urban centers of Charleroi and Mons (Figure 6).

### 3.3.1. Pre-Industrialization Heritage

The origins of the industrialized activities in Belgium find their earliest traces already back to the late Middle Ages, where industries prosper with some of the resulting production exported outside the Walloon territory, for instance Dinant, which was an important center for brass working, and focused on the raw materials trade from England and Germany. From the other side, in Nivelles and in the west of the Comté de Hainaut, the linen industry was experienced, as well. Furthermore, master ironworkers developed new processes with records back to 1320. During the 16th century (noticed as the 'metal working period'), the prosperity, especially in Wallonia continued, where one of the densest steel regions of the western Europe was established; during the same period, two hundred factories operated in Wallonia (Figure 7), while the coal production was increased considerably in Liège. At the same time, the textile industry began its expansion in the region of Verviers and new glass industrial activity was developed in Hainaut and part of Brabant. Later, during 17th century, the war struck all business activities, and with its end the manufacture of nails, gunpowder, weapons and coal export were increased, while the glass industry from 1650 and onwards was developed around Charleroi and Jumet [36].

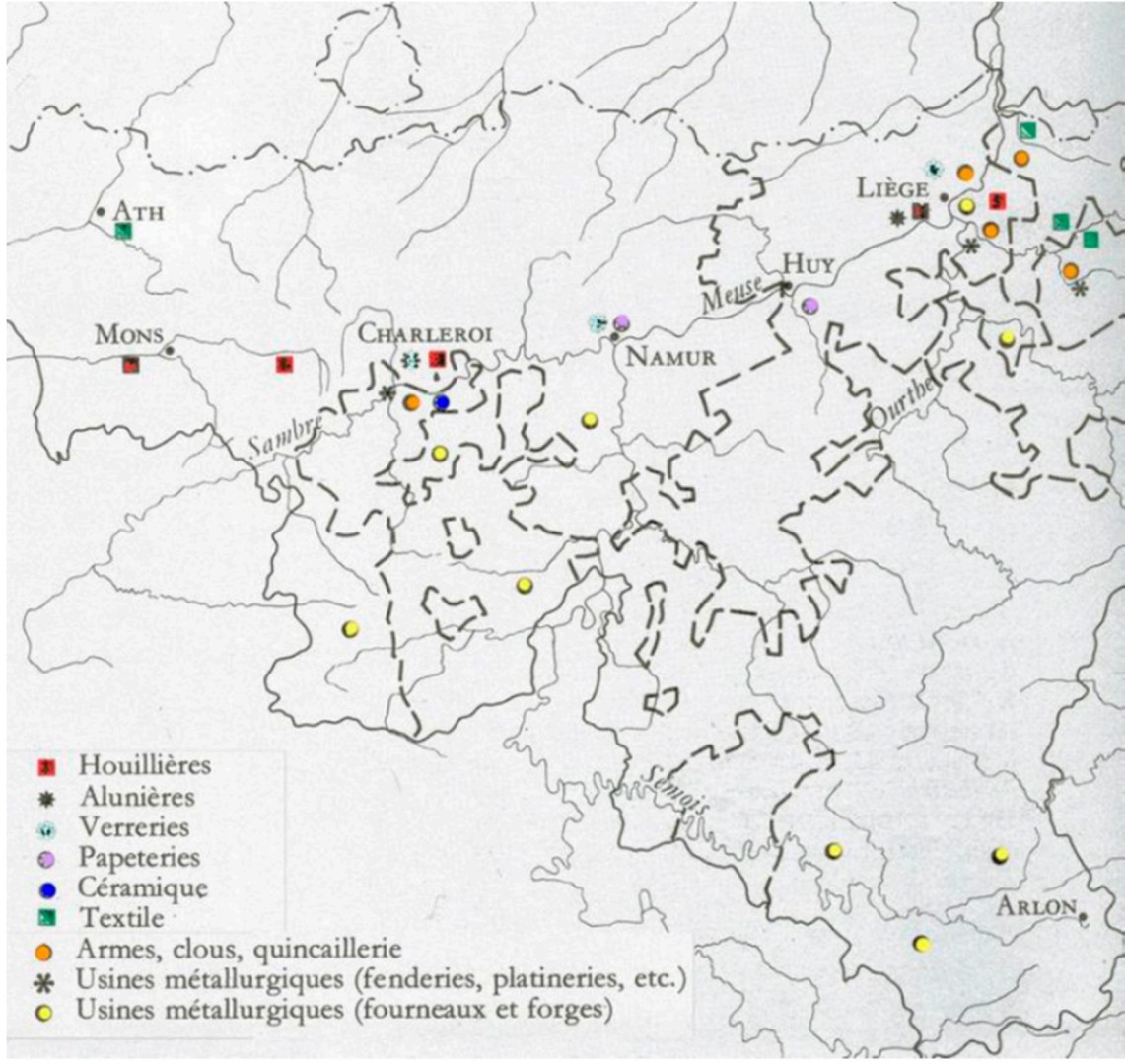

**Figure 7.** Geographical distribution of Walloon industries, 1680.

The earlier industry heritage was called pre-industrialization heritage, the representation in Belgium is the wool textile factory heritage [21,37]. During this period, the country has developed a vibrant trading tradition focused basically on the textile production, which flourished in Flanders and the iron processing in Wallonia. These key branches had been the prerequisites for the industrialization era, which followed. Indeed, Belgians maintained intensively the connection with Great Britain and in 1720 the first steam engine went into action near the region of Liège, a model inspired by T. Newcomen which used to draw out waste water from coal mining. This engine was succeeded later by another one around the areas of Mons and Charleroi boosting rapidly the coal and steel industrial activity. Later, in 1792, the country was dominated by Napoleon, who introduced the trade freedom and the market was flourished up in France [38].

In the middle ages, the regions of Flanders and Brabant were the centers of the production of wool fabrics, and the division of labor was very meticulous. After the 16th century, Belgium's cottage industry became more and more prosperous and became the real foundation of Belgium's industrial development in the 18th century, for instance, the museum of industry (Ghent) (Figure 8) [39].

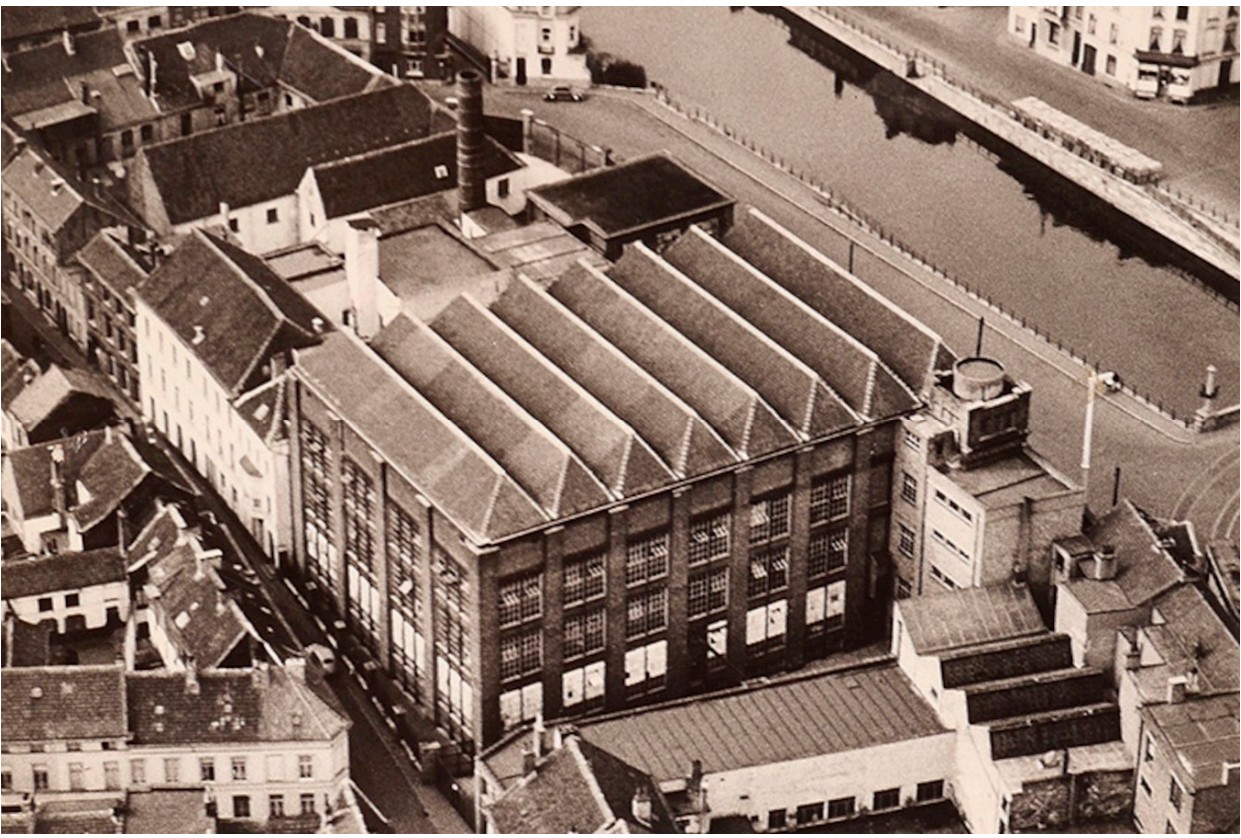

**Figure 8.** Aerial view of UCO Desmet-Guequier, 1948.

Because of its developed textile industry, Ghent developed from the end of the eighteenth century into an important center of the textile industry in Belgium [40]. During this period, the factory buildings were almost re-built from monasteries. Because the textile machine needs a bright and spacious space for running.

3.3.2. Industrialized Period from 1802 to the 1890s

The year 1802 is generally regarded as the beginning of the modern industry in Belgium [42]. After that, trade volume and production capacity stimulated each other and entered a period of rapid growth. At the end of the 19s, Belgium appeared a numerous beer factories, giant bridges, transportation hubs, and so on, for instance, before the First World

War, Belgium had more than 3000 beer factories [41]. Nonetheless, due to the large-scale expansion in this period, the design and construction of industrial buildings had not gone through too much serious thinking and deliberation. Almost no one studies what materials should be used and what forms should be adopted for these buildings. Therefore, in this period, the industrial buildings had a similar appearance to general public buildings [42].

After the Belgian Independence Movement. The Belgian government attempted to raise the level for Antwerp in international cross-border trade, compared with Rotterdam and Hamburger [43,44]. They built several transportation connection facilities with other central industrial cities in Antwerp. Established a sea transportation base and raw material import center for Wallonie's heavy industry.

According to this condition, Antwerp has a large number of industrial constructions, including warehouses, canals, railways, shipyards, dockyards, and so on. In the first half of the 20th century, two World Wars broke out, which objectively promoted the rapid discrepancy transformation of the Belgian industry between Wallonie and Flanders [45]. After the end of this period, the Belgian industry building got a wide range of influence from internationalization. It began to try to use new building materials, including prestressed reinforced concrete, steel, glass, and so on. The style of industrial buildings and the use of materials began to connect with the world.

## 4. Conservation and Renewal of the Industrial Heritage and Its Importance towards the Cultural Regeneration

An important reflection of the industrial heritage is its conservation and the strategies towards its renewal. In the protection of the transportation facilities, a few of them were redeveloped to experiential museums for the public. To ensure the integrity and authenticity of the railway heritage, and to keep it in a stable environment without over modernization. The care for the conservation and the capitalization of the cultural resources indicates a given society's degree of interest and an important connection with its past and history. Urban regeneration of former industrialized areas is also meant to avoid derelict areas and to promote strategies towards their conversion and exploitation aiming to display them as patrimony assets. Arguably, the conversion of industrial sites is a complex and an intangible aspect, which demands knowledge, expertise and understanding of the efficient strategies of heritage conservation towards the sustainable revitalisation of these sites. It is important to study the perspectives of the reconversion and the reuse of the formerly industrialized structures obtaining a new and multi-dimensional role as a source of economic stability and unique cultural identity. In this section, we provide an overview of interesting reconversion stories with a particular focus on our case-study of Belgium.

### 4.1. An European Overview of Interesting Examples

As we already explained, during the decade of '70s the continuous and rigid decline of the industrial activities in many European countries left a significant number of buildings, sites and former working places abandoned. Nonetheless, the idea towards their urbanization renewal has developed across Europe recently aiming to improve their living conditions and redevelop them through a gentrification process.

In this sub-section, some emblematic projects of famous rejuvenations around Europe are discussed [46]:

- Matadero (Madrid, Spain): formerly a slaughter house and meat market inaugurated in 1924 and it remains an interesting architectural reference of industrial's site transformation in Spain along with a renovation process towards its transformation into parks and promenades. Nowadays, Matadero is dedicated basically to art, design and literature and it accommodates exhibitions of a great variety.
- Wuk (Vienna, Spain): former locomotive factory (in 1855), turned into a cultural place (beginning of the 80s) promoting basically cultural activities with a variety of activities dedicated to contemporary art, etc.

- Kaapeli (Helsinki, Finland): former cable factory and a production site for Nokia Corporation. Today, the site has the function of a multimodal cultural center and accommodates three museums, twelve galleries, theatres, arts schools and other artistic disciplines.

Other interesting study areas of similar cases, such as the mining-related heritage tourism are found in Germany (Mining Museum with approximately 40,000 visitors); in Austria (Mine at Eisenerz opened in 1986 operated by a local company accommodates around 80,000 visitors per year) or in Italy the Talc and Graphine mine (Scopriminiera, 70 km from Turin) which is transformed into a museum, as well. Other examples of European cases and their renewal are found in textile industry heritage, for instance the Audax Museum in the Netherlands, which became a 'bridge' of the past textile production, the present and the future and receives many visitors throughout the year; or even the railway heritage tourism, which is quite widespread, especially for countries, such as the England [47].

### 4.2. A Focus in the Case of Belgium

In Belgium, not only locomotives and carriages, but also railways and stations were got a reservation, for instance, the Tramway Historique Lobbes-Thuin railway heritage which is one of the most aging tourism railways in Western Europe. Managing by ASVI. It includes trains, railways, stations and service facilities. Tourists to the line can inspect the vehicles not currently in use in a large showroom (Figure 9) [48].

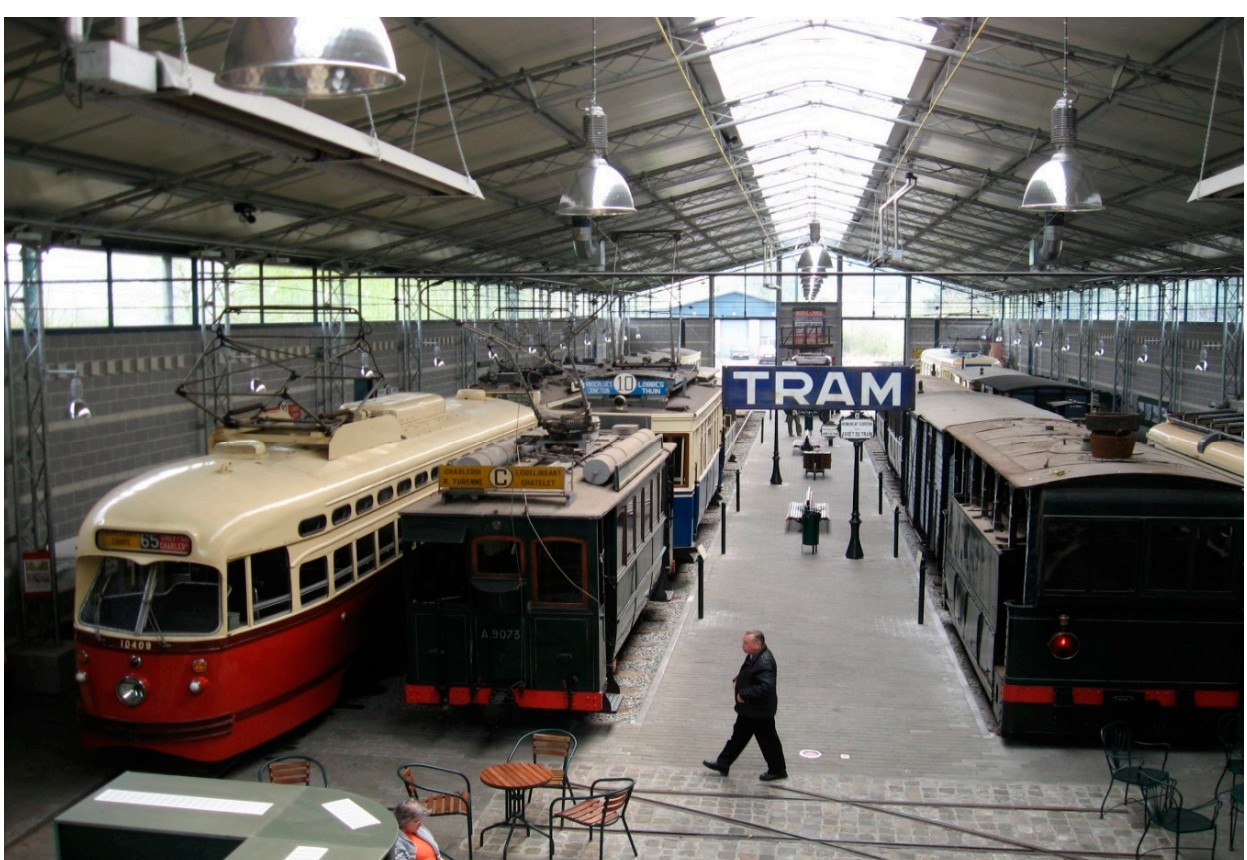

**Figure 9.** Showroom for vehicles display.

They can also take an old train through picturesque wooded countryside, passing the notable churches and terraced gardens and a belfry at Thuin (Figure 10) [48].

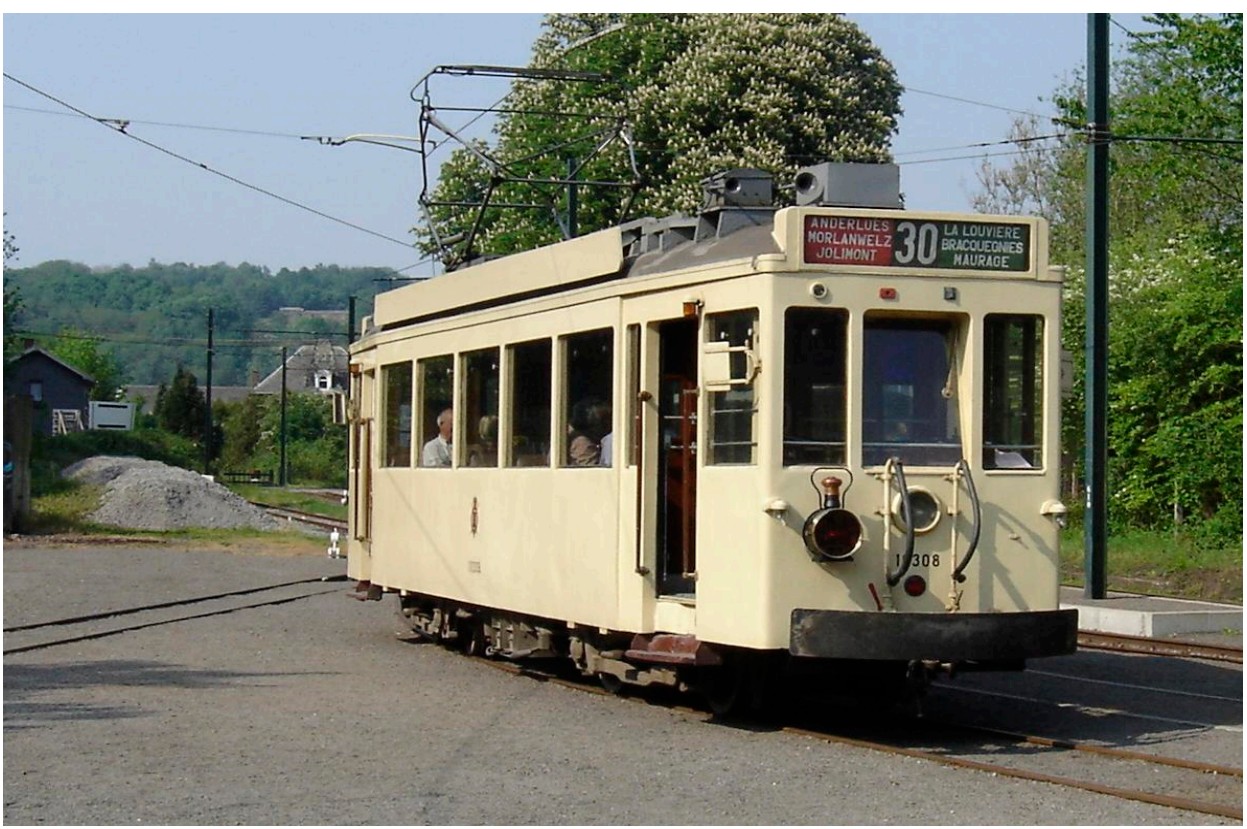

**Figure 10.** Train experience.

When talking about industrial heritage in Belgium, the mining heritage is an inextricable part. In this study, Wallonie is the representative case in Belgium. In 1720, a new air pressure pump was installed in the Liege coal mine area for the first time [49]. In 1814, Mons-Charleroi had an extraordinarily burgeoning in mining with more than 400 mining areas. The annual output had reached 1 million tons. This situation brought oceans of mining heritages after the industry-transforming [50]. Nonetheless, due to its huge area of land, and the lack of decoration outside, this phenomenon led to the rejection of the people for a long period. A typical mining heritage influences people is even more than that of a medieval church because these mining played an important role in promoting social development and economic prosperity.

Instead of demolishing these heritages, it is better to transform and renewal them, retain the imprint of industrial development, and make them continue to serve modern life. Therefore, the Belgian society reusing and conserving them selectively and gradually. Some of their pits have formed their microclimate, and many are rich in animal and plant life [51]. It is a route to preserve and develop ecotourism projects. Tourists can explore their unique landscape and eco-system while climbing the pit, and also experience the industry history in Belgium by tourists themselves [52]. Such as the project the trail of the pit in Walloon region, a kind of reusing in eco-tourism for industrial heritage, which became very popular in Belgium [53].

Complementary to the common status of the conservation brownfield, the reconstruction of the ecological environment. There are also challenging and bold attempts. A large number of mines, tunnels, factories, and industrial facilities have become the material evidence for reserving the history of Belgium, for instance, the Blegny-Mine world heritage site museum which has been listing in the UNESCO world heritage sites (Figure 11) [54,55].

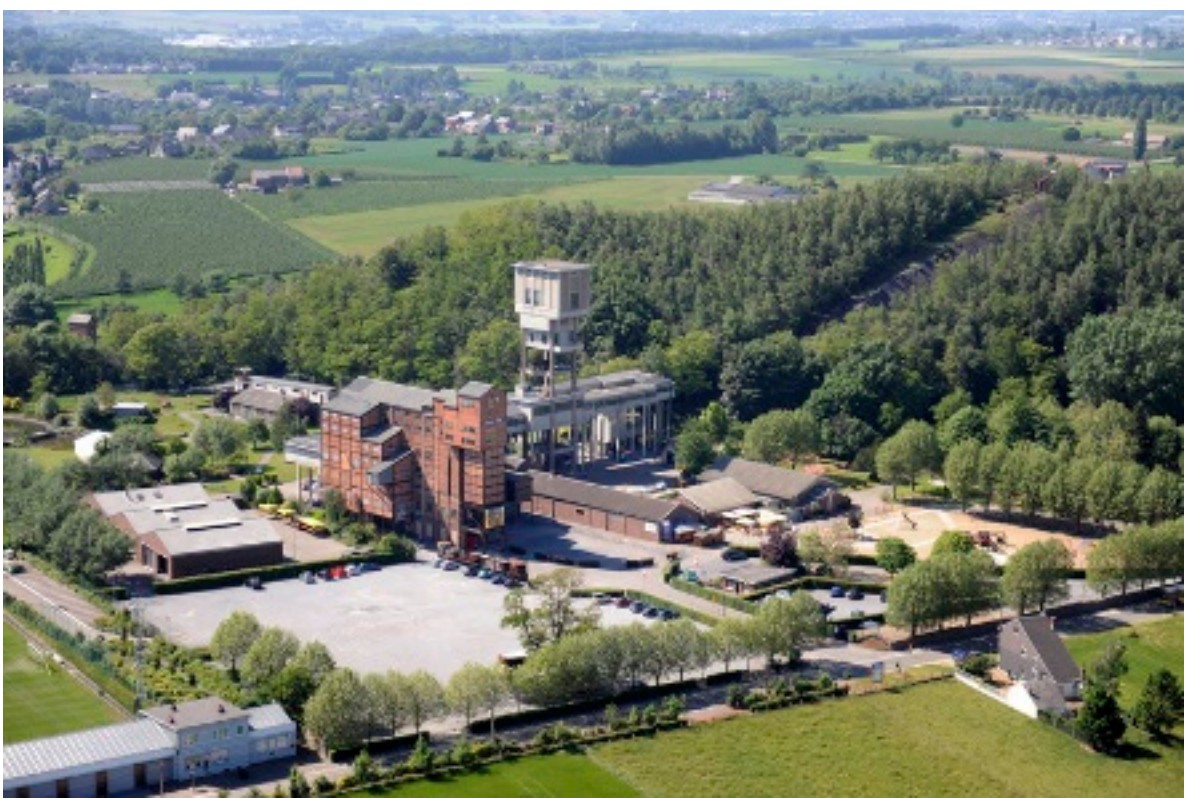

**Figure 11.** Aerial view of Blegny-Mine.

When it closed in 1980, the Blegny-Mine was the oldest and last remaining mine in the Liege district. The famous feature of that is tourists can experience the whole mining progress by old facilities. Like how workers did before 200 years. The museum has an intact mining assembly line. Tourists can go underground by a cage (Figure 12) [55].

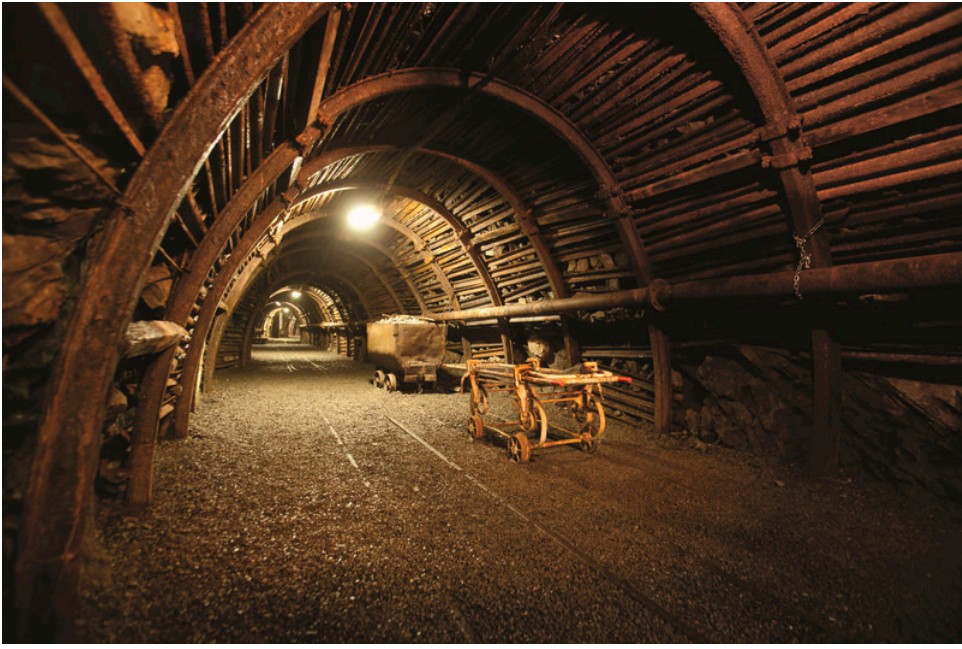

**Figure 12.** The underground operation area of Blegny-Mine.

Understanding the whole mining process by various forms of tour introductions, for instance, infrared-guided audio tours, lighting effects, animations, authentic reconstruc-

tions, and an exciting media show that stages the origin of the coal. The museum has multifaction facilities, tourists can not only take a scientific tour but also can take a walk on the stone heap, playgrounds, a petting zoo, a mini-train, or even a boat trip to Liege. It offers various tourism choices for families. As one of the 4 major mining sites of Wallonia recognized the UNESCO world heritage, and a site of regional industrial route "Route of Fire", Blegny-Mine plays an important role in the industrial heritage system in Belgium.

Belgium's industrial heritage protection takes various types of industrial museums as an important way of protection, preserving the original state of industrialization, reflecting and inheriting the achievements of industrial civilization. As a new form of special tourism with historical nostalgia and popular science experience as the main motivation and the integration of industry and tourism, industrial tourism in Belgium has not developed for a long time and has the smallest area of territory. However, compared with the traditional industrial tourism powers surrounding, Belgium has the highest density of industrial heritage tourism projects (Table 5).

**Table 5.** Industrial tourism case density ranking of Belgium's surrounding countries in ERIH.

| Ranking | Country | Case | Area of the Territory (km$^2$) | Case Density (Case/km$^2$) |
|---------|---------|------|-------------------------------|----------------------------|
| 1 | Belgium | 68 | 30,528 | 0.00222 |
| 2 | The Netherlands | 66 | 41,865 | 0.00157 |
| 3 | The U.K. | 380 | 244,100 | 0.00155 |
| 4 | Germany | 375 | 357,022 | 0.00105 |
| 5 | Portugal | 37 | 92,212 | 0.00040 |
| 6 | Italy | 105 | 301,333 | 0.00034 |
| 7 | France | 146 | 672,834 | 0.00021 |
| 8 | Spain | 102 | 506,000 | 0.00020 |

## 5. Main Findings and Perspectives

The modern industry in Belgium was developed in 1802. At the end of the 19th century, Belgium, a small country, has become one of the big capitalist economies, the second-largest economy in the world behind England. Due to its special national conditions and history, the industry developing and reservation has its outstanding routes and characteristics.

- Firstly, the structure of industrial heritage is very clear, each region has its own prominent industrial structure system. In the Walloon region the heavy industrialization and Flanders the textile industry. In Brussels, its service industry. The protection and research of each region are relatively independent. This feature enables each region of Belgium to play its own industrial advantages, resource-intensive, and create the greatest value. At the same time, due to the geographical proximity of each region, the logistics time and cost between them are greatly reduced. They are relatively independent but interdependent. Meanwhile, from the perspective of the whole country, the industrial chain of Belgium is complete and balanced. This is crucial to the development of industry.

- Secondly, despite the Industrial Revolution in Belgium was created followed by the U.K., but there was no leading sector in Belgium that was similar to the textile industry in the U.K. The basic industry in the Walloon region had a much greater decisive role in the change of industrial structure. Therefore, the kinds of industrial heritages in Belgium are mix and balance.

- Thirdly, each successful protection case is protected and updated according to its actual conditions. As a small country in Western Europe, due to the particularity of geopolitics, each region has its characteristics. The industrial allocation and market of Belgium are greatly influenced by the traditional powers around Belgium. The main industry affected by the U.K. in the northwest is the textile industry, the main industry affected by the Netherlands in the northern region is the logistics industry, the main

industry affected by Germany in the eastern region is the manufacturing industry, and the main industry affected by France in the southwest region is the mining industry.
- Fourthly, the industrial tourism in Belgium is developed. As mentioned in Section 4, Wallonie has a series of successful cases. From the train museum to the mining park. The industrial tourism not only promotes the development of the surrounding economy but also well preserve the cultural imprint of Belgium's industrial age, as a vector for cultural regeneration.

Belgium's industrial heritage cannot be compared with the traditional Western European industrial powers in many aspects. It is not as old as the U.K in terms of period and industrial categories, nor as grand as Germany in terms of land occupation scale, but it still has its outstanding regional characteristics, for instance, it started to develop very early (Figure 13) [56].

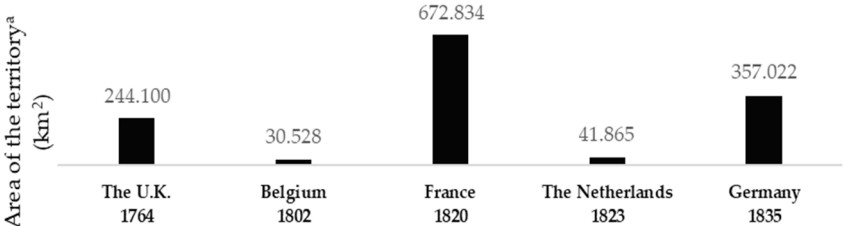

**Figure 13.** Comparative analysis with main industrial countries around Belgium ([a]. National data derived from national statistical records, see Appendix A Table A3. [b]. The date of beginning from Western European countries is based on the data derived from Hudson, Kenneth (1979).).

Heavy industry promotion leads to a flourishing development for transportation. Especially on railway system establishment [57]. In 1875, the overall railway length had been approached 3000 km. After that, Belgium has the densest railway system in Europe (Figure 14) [58].

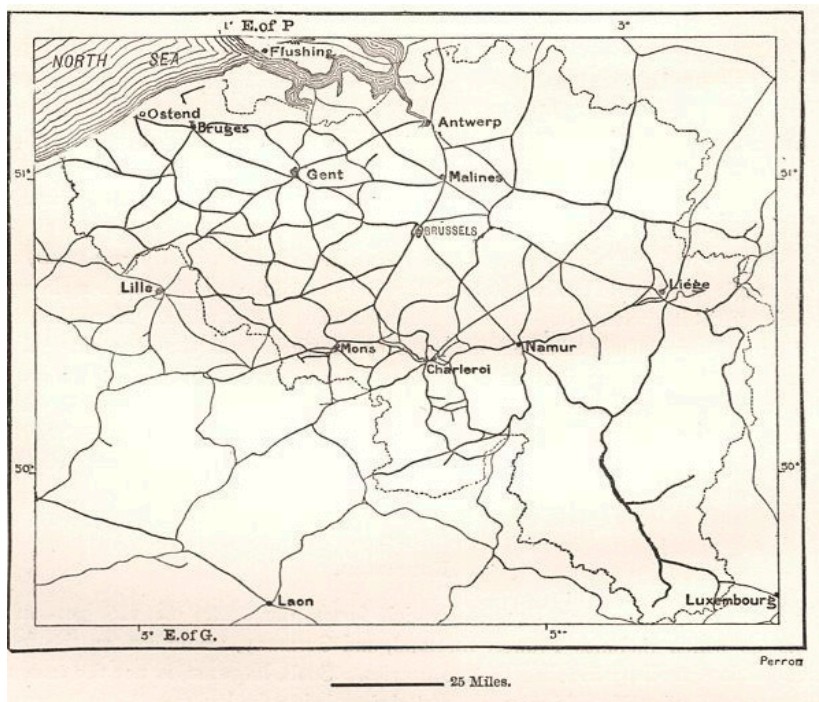

**Figure 14.** Railway Map of Belgium. Sketch map 1885 old antique vintage plan chart.

Therefore, railway facilities could be one of the representative industry heritages of Belgium (Figure 15) which are evolving and developing ceaselessly. In the beginning, most lines were originally steam-worked, but many were electrified and in later years others were worked by diesel traction. Most lines were laid alongside roads, and some carried freight as well as passengers.

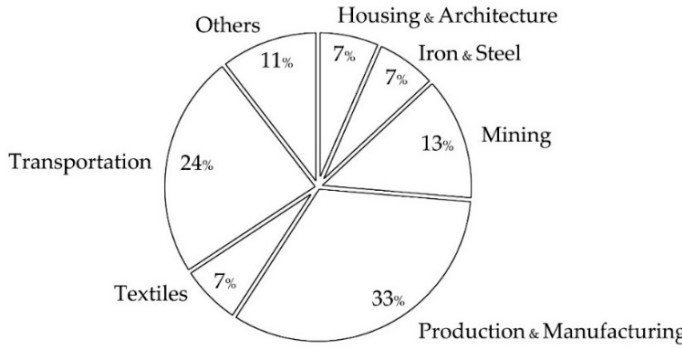

**Figure 15.** Categories of Belgium by the industrial heritage in ERIH, 2020.

At present, some of the abandoned railway heritages were re-used to scenic spots for industrial tourism. Each year, it attracts thousands of travelers all around the world. Besides, even though the territory of Belgium is limited, but not only railway transportation is developed, but also canal transportation. It has several industrial monuments of the highest quality. For example, the Strépy-thieu boat lift on the canal du center was added to the world heritage list of UNESCO world heritage sites in 1998 [59].

Besides, the port of Antwerp and the port of Liege are two of the most active ports in continental Europe [60]. There are several facilities for the port logistics industry, such as warehouse, dock, navigation lock, and so on. These large numbers of industrial heritages, witnessing the enlightenment, development, and expansion of Belgium's modern industry (Table 6).

**Table 6.** The representative Belgium industrial heritage sites in transportation in ERIH.

| Name | City | Date of Construction | Category |
| --- | --- | --- | --- |
| Central Station | Antwerp | 1886 | Railway station |
| Tour et Taxis | Brussels | 1907 | Transport center |
| The Strépy-Thieu Giant Boat Lift | Le Roeulx | 1950 | Boatlift |
| Ronquieres inclined plane | Ronquieres | 1960 | Inclined plane |
| Tram museum | Schepdaal | 1962 | Tram museum |
| Canal du Centre | Thieu | 1885 | Boatlift |
| Tramway Historique Lobbes-Thuin | Thieu | 1890 | Tramway |

Integrating conservation and management techniques towards the revitalization of industrial heritage is a key factor for cultural regeneration and a liaison between the past and the present, which requires, nonetheless, a general consensus of all the included parties and stakeholders and a know how to reveal their cultural significance for future generations. The understanding of the particular characteristics of each site and the data collection is always the primordial step towards this direction, which needs an interdisciplinary and analytical research work regarding the definition of the aesthetic, historical, scientific and social value of these sites. Secondly, the process should include an efficient management of the collected data, which will be the key for policy development with the identification of the physical constraints and opportunities for the reconversion of each site compiling all the stakeholders involved, who will develop the reconversion plan and the relevant policies.

## 6. Conclusions

As the second traditional industrial country to start the industrial revolution in the world. without a doubt, Belgium has oceans of successful industrial heritage projects and protection experience. However, there is not a comprehensive overview of its conservation and renewal. This research developed and analyzed its developing routine and current status. The renewal of the industrial heritage in Belgium shows the characteristics of aggregation, regionalism, a high degree of completion.

According to the literature review of the industrial heritage, this research puts industrial site, industrial land, brownfield, industrial heritage together, certified their conceptions, internal relationships, and straighten out the mutual logically them, which can help researchers to process further and similar research in the future. Due to its limited territory and relatively single industry structure, its types and new projects of industrial heritage protection in Belgium do not abound as other industrial countries. From outside, this research has overviewed the conservation and renewal of industrial heritage in Belgium, has not put into its neighborhood countries even world-wide. From inside, this research was not overviewed on each kind of specific industrial projects (industrial land, industrial site, brownfield).

In further researches, researchers can put the research into a broader range of scope, not only aims in an individual country. The overview of the brownfields, existing industrial heritage should have a general survey and database collection. Along with the conservation and renewal of the industrial heritage processing, the range of this research can be the border and deeper by including structural and cohesion policies of the successful stories of the past. The European experiences reveal the importance and the added value of the regeneration of the past industrial heritage and its impacts to the urban, economic and social environment of the territories (i.e., tourism, infrastructures/services, etc.). Nonetheless, it is crucially important, especially for countries with a long industrial history, as Belgium, to develop accurate databases and inventories of these sites and identify the opportunities and constraints. A key recommendation for further exploit the existing heritage is to focus on related policies (locally) and provide effective governance systems aiming to enable partnerships and networking. Adapting the experiences of the past and to improve the expertise and the know-how is always a successful mechanism to disseminate best practices and guide city actors. Eventual research areas are suggested to be developed including a detailed assessment of the existing industrial heritage (inventories), evaluating, mapping and exploring the sites. For decision-makers, Belgium's industrial heritage needs to form the scale benefit of regional integration, closely combine with tourism reuse, increase the way of utilization, enhance the degree of specialization, and emphasize the understanding and dissemination of industrial civilization.

From the experience of Belgium, generally, industrial heritage not only does not become a burden of urban development but also becomes the stage of people's creative life, even as a vector for cultural regeneration, this is depending on the located policies. The main difference between Belgian industrial heritage and other traditional heritage lies in its value of renewal, that is, to achieve the purpose of "protection" through "renewal". In this research, the conservation and renewal of industrial heritage in Belgium show oceans of experience that we can learn for further protection.

- Protection does not necessarily require large-scale demolition and reconstruction, and maintaining the status quo is also a protection mode.
- Although the initial stage of industrial development was at the expense of the natural environment. if it was protected by scientific routes. The brownfield can restore an eco-friendly environment.
- Through the protection of industrial heritage, it is also the protection of traditional culture and social memory.
- The Belgian government and research paid deep attention to science education for children and brownfield community reconstruction.

- Industrial heritage tourism is very developed in Belgium, there are great numbers of types of choices for tourists.
- The current situation of Belgian industrial heritage is consistent with the research trend of academic research, which highlights the guiding significance of the leading academic to the actual industry.
- Despite all this, due to its limited territory of land and separated regional situation. The development of industrial heritage has challenges, for instance, limited categories, lack of driving force for development, social and economic development slows down.

**Author Contributions:** J.Z. and J.C. developed the research topic. J.Z. prepared the writing-original draft. S.K. charged the writing-review and editing. The project administration was by V.B. All the authors contributed to the paper editing. All authors have read and agreed to the published version of the manuscript.

**Funding:** This research received no external funding.

**Informed Consent Statement:** Informed consent was obtained from all subjects involved in the study.

**Acknowledgments:** In this research, thanks to the knowledge officer library & collection of MIAT and the commercial department of Blegny-Mine. This research was funded and supported by the CoMod (Compacité urbaine sous l'angle de la modélisation mathématique (théorie des graphes et des jeux) project.

**Conflicts of Interest:** The authors declare no conflict of interest.

## Abbreviations

| Acronym/Abbreviation | Explanation |
| --- | --- |
| TICCIH | The International Committee for the Conservation of the Industrial Heritage |
| MIAT | Dutch: Museum over industrie, arbeid en textiel |
| VVIA | Dutch: Vlaamse Vereeniging voor Industriële Archeologie |
| UNESCO | The United Nations Educational, Scientific and Cultural Organization |
| UCO | French: Union Cotonnière |
| PIWB | French: Patrimoine Industriel Wallonie-Bruxelles |
| ASVI | French: Association pour la Sauvegarde du Chemin de Fer Vicinal á Thuin |
| ERIH | European Route of Industrial Heritage |

### Appendix A

**Table A1.** Schedule of main industrial heritage inspection and research institutions in Western European countries.

| Abbreviation | Online Resources |
| --- | --- |
| AIA | https://industrial-archaeology.org/ |
| TICCIH Deutsch | https://ticcih.org/germany/ |
| CILAC | https://www.cilac.com/ |
| FIEN | https://www.industrieel-erfgoed.nl/ |

**Table A2.** Names and types of Industrial heritages' in Wallonie.

| No. | Name | Type |
| --- | --- | --- |
| 1 | Tour Saint Albert in People-lez-Binche | Coaling |
| 2 | Offices of the Charbonnage du Bois d'Avroy in Liege | - |
| 3 | Blegny-Mine slagging | - |
| 4 | Marcasse coal mine in Wasmes | - |
| 5 | Well N ° 2 of the Anderlues coal mine | - |
| 6 | Decanter cycle of the coal mining of the Fief de Lamberchies in Quaregnon | - |
| 7 | Charbonnages du Gouffre N ° 10 in Châtelineau | - |
| 8 | Charbonnage du Levant in Cuesmes | - |
| 9 | Charbonnage de la Forte-Taille in Montigny-le-Tilleul | - |

**Table A2.** *Cont.*

| No. | Name | Type |
|---|---|---|
| 10 | Dumont-Wauthier lime kiln and circular kiln in Amay | - |
| 11 | Dapsens cement plant in Vaulx | - |
| 12 | Lessines boat loader | - |
| 13 | Martelange slate quarry | - |
| 14 | Former Carsid steel site | Metallurgy |
| 15 | HFB in Ougrée | - |
| 16 | Forges de Clabecq | - |
| 17 | Offices of the Leonard-Gito Factory in Marchiennes-au-Pont | - |
| 18 | Saint-Eloi rolling mill in Thy-le-Château | - |
| 19 | ACEC factory in Herstal | Mechanical, Metal, Electrical and Electronic Construction |
| 20 | Boël factory in La Louvière | - |
| 21 | Kinkempois workshop in Liege | Construction, Servicing And Maintenance of Railway Equipment |
| 22 | Dubois Pottery in Bouffioulx | Pottery |
| 23 | Mirox (Miroiterie) in Marchienne-au-Pont | Glass |
| 24 | Glaverbel in Roux | - |
| 25 | Vanoutryve factory in Mouscron | Textile |
| 26 | La Vesdre factory in Dison | - |
| 27 | Defosses & Fils wool wash house in Dolhain | - |
| 28 | Despa factory in Theux | - |
| 29 | Toulemonde-Destombes spinning mill in Dottignies | - |
| 30 | Bonneterie Dujardin in Leuze-en-Hainaut | - |
| 31 | Le Bon Grain in Morlanwelz | Boulangery, Pastry, Biscuitery, Food Pasta |
| 32 | Brasserie des Alliés in Marchienne-au-Pont | Brassery |
| 33 | Union Brasserie-Malterie in Jumet | - |
| 34 | The Wez plant (Carsid) in Marcinelle | Power Station |
| 35 | The Marchienne Energy Plant | - |
| 36 | The Cockerill SA power station in Seraing | - |
| 37 | The Monceau power station | - |
| 38 | N ° 1 in Seraing | Pumping Station |
| 39 | City of the gunsmiths in Liege | City, Patron House |
| 40 | Bonnier barracks in Grâce-Hollogne | - |
| 41 | Coron of Archies | - |
| 42 | Casino Salvay in Couillet | - |
| 43 | Château Saroléa in Cheratte | - |
| 44 | Hôtel de Biolley in Verviers | - |
| 45 | Château Industriel Léonard-Giot in Marchienne-au-Pont | - |
| 46 | Château Pirmez in Gilly | - |
| 47 | Château Baudoux in Jumet | - |

**Table A3.** Main industrial countries around Belgium.

| Country | Reference |
|---|---|
| The U.K. | Demographic Yearbook—Table 3: Population by sex, rate of population increase, surface area, and density. United Nations Statistics Division. 2012. Retrieved 9 August 2015. |
| Belgium | "be.STAT". Bestat.statbel.fgov.be. 26 November 2019. |
| France | Economic, social and territorial situation of France—La Réunion |
| The Netherlands | Netherlands Country/Territory Profile ǀ CAPA |
| Germany | "Germany". CIA World Factbook. Archived from the original on 11 February 2016. Retrieved 29 March 2020. |
| Italy | Italy: total area of by region 2017 ǀ Statista |
| Spain | Spain Country/Territory Profile ǀ CAPA |
| Portugal | Economic, social and territorial situation of Portugal |

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
