# Peer review of "The Overview of the Conservation and Renewal of the Industrial Belgian Heritage as a Vector for Cultural Regeneration"

_information, doi:10.3390/info12010027_

Round 1

Reviewer 1 Report

The manuscript title "The Overview of the Conservation and Renewal of the Industrial Belgian Heritage as a Vector for Cultural Regeneration" is well written. This manuscript definitely provides information that is interesting for the reader of information. I do not find anything that is against the publication of the manuscript. I recommend accepting the manuscript in the present form. I congratulate author fora great piece of work.

Author Response

Dear Editor,

Dear Reviewers,

With this cover letter, we will submit the revised Manuscript (information-1021642) entitled ‘The overview of the conservation and renewal of the industrial Belgian heritage as a vector for cultural regeneration’ for publication in the journal ‘Information’. We would like to thank you for the careful and constructive reviews and suggestions. Based on the comments received, we have revised the manuscript and updated its current version.

Editor

It would be appreciated if you rephrase the content to avoid paragraph copy.

The similarity report is attached. Please rephrase the highlighted parts (especially in the abstract, analysis, and conclusion part) in your own words. You can keep the equations and basic concepts as it is.

Reply to Editor’s comments:

Dear Editor,

Thank you for your comment and suggestion, which are well considered at the updated version of our submitted manuscript. An exhaustive revision of each sentence of the manuscript was performed by rephrasing accordingly its context following the guide of your report proposed aiming to reduce the similarity index. The proposed changes are presented to the updated version of the manuscript. All the modifications following your comments and suggestions are included at the submitted version with ‘green’ highlights.

Reviewer 1:

The manuscript title "The Overview of the Conservation and Renewal of the Industrial Belgian Heritage as a Vector for Cultural Regeneration" is well written. This manuscript definitely provides information that is interesting for the reader of information. I do not find anything that is against the publication of the manuscript. I recommend accepting the manuscript in the present form. I congratulate author fora great piece of work.

Reply:

Dear Reviewer,

Thank you for your comments and your time to review our manuscript.

Reviewer 2 Report

The submitted work aims to illuminate the correlation between the conservation and renewal of industrial Belgian heritage with the cultural and economical regeneration.

The selected subject of research is quite important, as the conservation of industrial heritage is a burning issue for sustainable development along with the appropriate respect to history. This is the reason why the ICOMOS (International Council on Monuments and Sites) is establishing the International Scientific Committee on Industrial Heritage (ISCIH).

The authors correctly mention the International Committee for the protection of industrial heritage (TICCIH) but it is suggested to include the latest Joint ICOMOS – TICCIH Principles for the Conservation of Industrial Heritage, also called "The Dublin Principles", adopted in Paris in 2011.

As for the content of the submitted work, in the second page and lines 62-68 the authors mention the outline of their work but, in my opinion, it isn’t in full accordance with the content.

Taking in account that the paper is submitted as a review/overview, the chapters 1 and 2 are good in length and well informative. Chapter 3 (the overview of the industrial heritage in Belgium) fill the largest part of this work. This would be good if the discussion and conclusion sections (5 and 6) were proportionally large. This makes the work unbalanced. A very good coverage of the industrial heritage development should be followed by a sound analysis of the impacts. In this work we learn a lot for the industrial development in Belgium but there is a weak correlation with the impact on cultural regeneration.

Furthermore, the manuscript needs a thorough reading because of the several grammatical, vocabulary, punctuation and syntax errors probably due to the translation from French (e.g., line 361).  

In several cases a misplaced full stop punctuation mark (.) made reading harder as for example in lines 181, 194, 432, 477 and 560.

In lines 230-237 the text is hard to read and need check.

In line 235: “William, I wanted the south of his realm [..]” the authors probably want to mention the William I of the Netherlands.

Lines 431-433 need rephrase: “The year of 1802 is generally regarded as the beginning of the development of modern industry in Belgium [42]. After 20 years of the Industrial Revolution of the U.K. along with a drastic ascending in trade, industry productivity had a drastic ascending.”.

Also, in several cases, the in-text numbering of figures and table is missing e.g., lines 157, 162, 319, 439, 447, 465, 485, 490.  

Finally, I strongly suggest the resubmission after major revisions to a more suitable journal such as Heritage.

Author Response

Dear Editor,

Dear Reviewers,

With this cover letter, we will submit the revised Manuscript (information-1021642) entitled ‘The overview of the conservation and renewal of the industrial Belgian heritage as a vector for cultural regeneration’ for publication in the journal ‘Information’. We would like to thank you for the careful and constructive reviews and suggestions. Based on the comments received, we have revised the manuscript and updated its current version.

Editor

It would be appreciated if you rephrase the content to avoid paragraph copy.

The similarity report is attached. Please rephrase the highlighted parts (especially in the abstract, analysis, and conclusion part) in your own words. You can keep the equations and basic concepts as it is.

Reply to Editor’s comments:

Dear Editor,

Thank you for your comment and suggestion, which are well-considered in the updated version of our submitted manuscript. An exhaustive revision of each sentence of the manuscript was performed by rephrasing accordingly its context following the guide of your report proposed aiming to reduce the similarity index. The proposed changes are presented in the updated version of the manuscript. All the modifications following your comments and suggestions are included in the submitted version with ‘green’ highlights.

Reviewer 2:

The submitted work aims to illuminate the correlation between the conservation and renewal of industrial Belgian heritage with cultural and economical regeneration.

The selected subject of research is quite important, as the conservation of industrial heritage is a burning issue for sustainable development along with the appropriate respect to history. This is the reason why the ICOMOS (International Council on Monuments and Sites) is establishing the International Scientific Committee on Industrial Heritage (ISCIH).

The authors correctly mention the International Committee for the protection of industrial heritage (TICCIH) but it is suggested to include the latest Joint ICOMOS – TICCIH Principles for the Conservation of Industrial Heritage, also called "The Dublin Principles", adopted in Paris in 2011.

As for the content of the submitted work, on the second page and lines 62-68 the authors mention the outline of their work but, in my opinion, it isn’t in full accordance with the content.

Taking into account that the paper is submitted as a review/overview, chapters 1 and 2 are good in length and well informative. Chapter 3 (the overview of the industrial heritage in Belgium) fill the largest part of this work. This would be good if the discussion and conclusion sections (5 and 6) were proportionally large. This makes the work unbalanced. Very good coverage of the industrial heritage development should be followed by a sound analysis of the impacts. In this work, we learn a lot about industrial development in Belgium but there is a weak correlation with the impact on cultural regeneration.

Furthermore, the manuscript needs a thorough reading because of the several grammatical, vocabulary, punctuation, and syntax errors probably due to the translation from French (e.g., line 361).

In several cases, a misplaced full stop punctuation mark (.) made reading harder as for example in lines 181, 194, 432, 477 and 560.

In lines 230-237, the text is hard to read and needs check.

In line 235: “William, I wanted the south of his realm [..]” the authors probably want to mention the William I of the Netherlands.

Lines 431-433 need rephrase: “The year of 1802 is generally regarded as the beginning of the development of modern industry in Belgium [42]. After 20 years of the Industrial Revolution of the U.K. Along with a drastic ascending in trade, industry productivity had a drastic ascending.”.

Also, in several cases, the in-text numbering of figures and table is missing e.g., lines 157, 162, 319, 439, 447, 465, 485, 490.

Finally, I strongly suggest the resubmission after major revisions to a more suitable journal such as Heritage.

Answers:

Dear Reviewer,

First of all, we would like to thank you for your extensive review and your suggestions. We have considered carefully all your comments and the relevant modifications are included at the updated version of the manuscript submitted. All the modifications following your comments and suggestions are included at the submitted version with ‘red’ highlights.

In particular:

  1. It is suggested to include the latest Joint ICOMOS – TICCIH Principles for the Conservation of Industrial Heritage.

Reply:

Thank you so much for this suggestion. Indeed, the Dublin principle also has a significant meaning for industrial heritage protection. A presentation and analysis of this reference can improve the paper's timeliness and authority. This part has been added to chapter 2.3 based on your suggestion.

  1. As for the content of the submitted work, in the second page and lines 62-68 the authors mention the outline of their work but, in my opinion, it isn’t in full accordance with the content.

Reply:

We modified this part according to the structure and content of the paper, the current version has higher accordance with the content.

  1. This would be good if the discussion and conclusion sections (5 and 6) were proportionally large. This makes the work unbalanced. Very good coverage of the industrial heritage development should be followed by a sound analysis of the impacts. In this work, we learn a lot about industrial development in Belgium but there is a weak correlation with the impact on cultural regeneration.

Reply:

We modified chapters 5 & 6 based on your suggestion, adding relevant content on how the industrial heritage development followed by an analysis of the impact. Enhance the correlation with the impact on cultural regeneration.

  1. The manuscript needs a thorough reading because of the several grammatical, vocabulary, punctuation and syntax errors probably due to the translation from French (e.g., line 361).  

Reply:

We double-checked the grammatical, vocabulary, punctuation, syntax, and French translation errors revised them correctly.

  1. In several cases, a misplaced full stop punctuation mark (.) made reading harder as for example in lines 181, 194, 432, 477 and 560.

Reply:

Sorry for this reading experience. All the punctuation marks you proposed have been modified. Then, we checked the punctuation completely again. Thank you for this reminder.

  1. In lines 230-237 the text is hard to read and need check. In line 235: “William, I wanted the south of his realm [..]” the authors probably want to mention the William I of the Netherlands.

Reply:

We feel sorry about this hard reading experience. The lines 230-237, including 235 have been embellished, the sentence has logistic mistakes has been deleted, and the structure of sentences has been adjusted. Hope the content can be read smoother than before.

  1. Lines 431-433 need rephrase:

Reply:

Thank you for this comment, the old lines 431-433 has been rephrased totally. For updates, we chose more precise and proper words.

  1. Also, in several cases the in-text numbering of figures and table is missing e.g., lines 157, 162, 319, 439, 447, 465, 485, 490.

Reply:

This article was written by several of us. When the work is handed over many times, there may be some mistakes due to negligence or software version problems. In any case, we are sorry for this mistake, all the numbers of figures and tables have now been checked and modified.

  1. Finally, I strongly suggest the resubmission after major revisions to a more suitable journal such as Heritage.

Reply:

Thank you for this suggestion. To be honest, our group had a discussion about this issue before submitting it, the same question. ‘Heritage’ is a perfect choice, without a doubt. However, after comprehensive consideration, according to our work arrangement, we finally decided choice ‘Information’ for this paper.

Reviewer 3 Report

The article aims to identify and classify the cultural industrial Belgian heritage and the various actions carried out for its conservation and renewal. The work of data collection and analysis is accurate and exhaustive, as evidenced by the rich consulted bibliography.

The distinction between industrial site and industrial heritage is very interesting as well as the idea that a true renewal of industrial heritage can only arise from the enhancement of the surrounding landscape, highlighting the transformations that the industrialization has brought to the territory and society.

Starting from these concepts, the authors correctly consider Industrial Belgian Heritage also all the infrastructures necessary for its proper development, such as storage sites, quarries, infrastructure for workers and, above all, railways, canals, ports and roads.

The authors repeat several times (perhaps too often) that Belgium was the first country involved in the Industrial Revolution after the UK. Various types of industries characterized the various regions of the country also in relation to specific traditions and the availability of raw materials. It is therefore evident that a correct conservation and renewal of industrial structures and sites is very important for the enhancement of the Belgian history and culture and for the development of sustainable tourism.  

in chapter 4 the concept of industrial heritage renewal could be examine in depth  by providing more detailed examples and comparing the results of the different experiences, identifying strong points and weaknesses. If available, it would be advisable to insert a table and discuss the data relating to the inflow of tourists in the various sites: how many visitors, what  age, origin, cultural level, etc. are most involved and why according to the authors.

In my opinion, the conclusions of this commendable data collection should be more incisive by proposing some concrete operative solutions to further increase the perception of the importance of industrial heritage at local and European level.

For what is regarding the exposition, I am not able to evaluate the English level, but it seems to me that some parts are a bit hasty and the sentences are extremely simplified. Some concepts are not completely clear to me (e.g. between lines 476-481) and there are some repetitions (e.g. 147, 552, 575): in general, it is necessary to revise the punctuation (sentences interrupted) to make the text clearer.

Some others remarks:

- the old photos and maps are very beautiful, but it is advisable to indicate their origin and, where missing, the date.

-the acronyms at the bottom of the article are not easy to consult: better to explicit them in the text (such as was done with UNESCO in line 328)

- often the number of the figure / table is missing, or it is wrong (157, 162, 275, 319, 423, 439, 447, 465, 469, 485, 490, 499, 533, 538, 542)

- line 235: remove the comma between William and I

- line 261: Cochlear or Koclear?

Author Response

Dear Editor,

Dear Reviewers,

With this cover letter, we will submit the revised Manuscript (information-1021642) entitled ‘The overview of the conservation and renewal of the industrial Belgian heritage as a vector for cultural regeneration’ for publication in the journal ‘Information’. We would like to thank you for the careful and constructive reviews and suggestions. Based on the comments received, we have revised the manuscript and updated its current version.

Editor

It would be appreciated if you rephrase the content to avoid paragraph copy.

The similarity report is attached. Please rephrase the highlighted parts (especially in the abstract, analysis, and conclusion part) in your own words. You can keep the equations and basic concepts as it is.

Reply to Editor’s comments:

Dear Editor,

Thank you for your comment and suggestion, which are well considered at the updated version of our submitted manuscript. An exhaustive revision of each sentence of the manuscript was performed by rephrasing accordingly its context following the guide of your report proposed aiming to reduce the similarity index. The proposed changes are presented to the updated version of the manuscript. All the modifications following your comments and suggestions are included at the submitted version with ‘green’ highlights.

Reviewer 3:

The article aims to identify and classify the cultural industrial Belgian heritage and the various actions carried out for its conservation and renewal. The work of data collection and analysis is accurate and exhaustive, as evidenced by the rich consulted bibliography.

The distinction between industrial site and industrial heritage is very interesting as well as the idea that a true renewal of industrial heritage can only arise from the enhancement of the surrounding landscape, highlighting the transformations that the industrialization has brought to the territory and society.

Starting from these concepts, the authors correctly consider Industrial Belgian Heritage also all the infrastructures necessary for its proper development, such as storage sites, quarries, infrastructure for workers and, above all, railways, canals, ports and roads.

The authors repeat several times (perhaps too often) that Belgium was the first country involved in the Industrial Revolution after the UK. Various types of industries characterized the various regions of the country also in relation to specific traditions and the availability of raw materials. It is therefore evident that a correct conservation and renewal of industrial structures and sites is very important for the enhancement of the Belgian history and culture and for the development of sustainable tourism.

In chapter 4 the concept of industrial heritage renewal could be examine in depth by providing more detailed examples and comparing the results of the different experiences, identifying strong points and weaknesses. If available, it would be advisable to insert a table and discuss the data relating to the inflow of tourists in the various sites: how many visitors, what age, origin, cultural level, etc. are most involved and why according to the authors.

In my opinion, the conclusions of this commendable data collection should be more incisive by proposing some concrete operative solutions to further increase the perception of the importance of industrial heritage at local and European level.

For what is regarding the exposition, I am not able to evaluate the English level, but it seems to me that some parts are a bit hasty and the sentences are extremely simplified. Some concepts are not completely clear to me (e.g. between lines 476-481) and there are some repetitions (e.g. 147, 552, 575): in general, it is necessary to revise the punctuation (sentences interrupted) to make the text clearer.

Some others remarks:

- the old photos and maps are very beautiful, but it is advisable to indicate their origin and, where missing, the date.

-the acronyms at the bottom of the article are not easy to consult: better to explicit them in the text (such as was done with UNESCO in line 328)

- often the number of the figure / table is missing, or it is wrong (157, 162, 275, 319, 423, 439, 447, 465, 469, 485, 490, 499, 533, 538, 542)

- line 235: remove the comma between William and I

- line 261: Cochlear or Koclear?

Answer:

Dear Reviewer,

First of all, we would like to thank you for your extensive review and your suggestions. We have considered carefully all your comments and the relevant modifications are included at the updated version of the manuscript submitted. All the modifications following your comments and suggestions are included at the submitted version with ‘purple’ highlights.

In particular:

  1. The authors repeat several times (perhaps too often) that Belgium was the first country involved in the Industrial Revolution after the UK. Various types of industries characterized the various regions of the country also in relation to specific traditions and the availability of raw materials. It is therefore evident that a correct conservation and renewal of industrial structures and sites is very important for the enhancement of the Belgian history and culture and for the development of sustainable tourism.

Reply: Indeed, we considered that explaining the fact of Belgium being the first country involved in the Industrial Revolution the core of this study. We totally agree on your point and observation that a correct conservation and renewal is necessary towards the achievement of the principles of sustainable tourism; nonetheless, the exploitation of the former industrialized sites in Belgium is quite low at the time being and further work (academic but also in practice) is requested towards this direction to enhance their value and propose strategies towards their rejuvenation. The advantages, as also explained in the manuscript, are numerous replying to the problematic of the lack of space and the parallel urban growth in modern cities, the opportunity of urban renewal and regeneration corresponding to the European strategies tackling the impacts of the climate change, etc. and undoubtably the development of a tourism, which will be developed in accordance with the principles of the sustainable development.

  1. In chapter 4 the concept of industrial heritage renewal could be examine in depth by providing more detailed examples and comparing the results of the different experiences, identifying strong points and weaknesses. If available, it would be advisable to insert a table and discuss the data relating to the inflow of tourists in the various sites: how many visitors, what age, origin, cultural level, etc. are most involved and why according to the authors.

Reply: Thank you for comment and interesting remark. We included at this chapter examples of European renewal and strategies of different cases to complete the review of the updated version of the submitted manuscript. Therefore, we modified the title of this chapter and we included two sub-chapters, from which the first summarizes interesting and representative examples of the scientific literature and the second part focusses on the case of Belgium (please check for more information the sub-sections 4.1 and 4.2) integrating the maximum of available information as recommended.

  1. In my opinion, the conclusions of this commendable data collection should be more incisive by proposing some concrete operative solutions to further increase the perception of the importance of industrial heritage at local and European level.

Reply: Thank you for the comment, which is considered accordingly at the revised version of the manuscript. Following your recommendations, we proposed and added some recommendations and solutions to increase the importance of the industrial heritage at local and European level (part of conclusions, highlighted accordingly).

  1. For what is regarding the exposition, I am not able to evaluate the English level, but it seems to me that some parts are a bit hasty and the sentences are extremely simplified. Some concepts are not completely clear to me (e.g. between lines 476-481) and there are some repetitions (e.g. 147, 552, 575): in general, it is necessary to revise the punctuation (sentences interrupted) to make the text clearer.

Reply:

  • Line 147: the authors removed the word ‘mines’ from the sentence to avoid the repetition with the ‘mining areas’ (see updated version of the manuscript)
  • Lines 476-481: the paragraph was revised and rephrased accordingly: ‘When talking about industrial heritage in Belgium, the mining heritage is an inextricable part. In this study, Wallonie is the representative case in Belgium. In 1720, a new air pressure pump was installed in the Liege coal mine area for the first time [50]. In 1814, Mons-Charleroi had an extraordinarily burgeoning in mining with more than 400 mining areas. The annual output had reached 1 million tons. This situation brought oceans of mining heritages after the industry-transforming [51]. Nonetheless, due to its huge area of land, and the lack of decoration outside, this phenomenon led to the rejection of the people for a long period. A typical mining heritage influences people is even more than that of a medieval church because these mining played an important role in promoting social development and economic prosperity’.
  • Line 552: the sentence is reformed and revised without interruptions accordingly: ‘Therefore, railway facilities could be one of the representative industry heritages of Belgium (Figure 19), which are evolving and developing ceaselessly’.
  • Lines 575-579: the paragraph was revised and corrected accordingly: ‘According to the literature review of the industrial heritage, this research puts industrial site, industrial land, brownfield, industrial heritage together, certified their conceptions, internal relationships, and straighten out the mutual logically them, which can help researchers to process further and similar research in the future’.

Some others remarks:

  1. The old photos and maps are very beautiful, but it is advisable to indicate their origin and, where missing, the date.

Reply: Thank you for your comment. Nonetheless, the information requested is cited and explained at the manuscript. The origins are provided when the Figures are referenced in the text and the date usually is noticed in the caption of each of the figures/tables proposed. Nonetheless, your comment is considered accordingly and we revised all the provided photos and maps as recommended (for instance Fig. 3, etc.). Figure 5 was removed from the updated version of the manuscript.

  1. The acronyms at the bottom of the article are not easy to consult: better to explicit them in the text (such as was done with UNESCO in line 328)

Reply: Thank you for your comment. Nonetheless, all the abbreviations are explained in detail at the end of the manuscript at the table of Nomenclature.

  1. Often the number of the figure / table is missing, or it is wrong (157, 162, 275, 319, 423, 439, 447, 465, 469, 485, 490, 499, 533, 538, 542)

Reply: Thank you for your comment. Indeed, there is a general problem with the numeration in the manuscript. The problem is taken under consideration and resolved at the updated version of the submitted manuscript; the figures and tables are cited in the text accordingly, as well.

  1. Line 235: remove the comma between William and I

Reply: Thank you for your comment. Indeed, it is William I, the remark is considered accordingly and corrected at the updated version of the manuscript.

  1. Line 261: Cochlear or Koclear?

Reply: Thank you for your comment. The name ‘Koclear’ is corrected accordingly at the updated version of the manuscript.

Reviewer 4 Report

Originality : Very low. This article could have been written twenty years ago. It evokes the current debates, but in a way too superficial. We would like to know how the experience acquired in the management of industrial heritage has changed the concept and its understanding on the one hand, and the actors involved on the other. From this point of view, the actors who took charge of the enhancement of the industrial heritage are absent from the analysis: public institutions, associations, architects, town planner, etc. More exactly, they are indicated, but without analysis of their role, and how it has evolved over the decades of experience mentioned.

Furthermore, the passage of successive phases: industrialization / deindustrialisation / cultural and tourist heritage of industrial sites seems to go without saying. It has not happened. It has been a very complicated way, humanely, economically, institutionally. Since the link betwwen industrial heritage and cultural regeneration, we would like to know what happened to the Pays de Liège, for example, which went through a huge depression due to deindustrialisation. 

Significance of Content : Good. Interesting. But there is a lack of references on this topic from other European countries, mainly Italy, Spain, Portugal (which have been also industrialized and are more touristic than Belgium), and from Germany, or more recently, Czech Republic. And : "Cultural regeneration" probably does not mean the same thing in the Pays de Liège and in Flanders ... Belgium is culturally marked by its historical division between Flemish and Walloons, and as the article suggests, but without going into depth, this division refers to a difference in economic history: in the 19th century, Wallonia had the primacy because of its very strong heavy industry ...

Quality of presentation : Excellent. 

Scientific Soudness : Low. The main flaw here is an inaccurate chronology of the industrialization phase, and an overly narrow reading of the pre-industrial phase. There is here a lack of historical reference on the question. To go quickly: the Industrial Revolution concerns the United Kingdom, and not just England (Britain), and in 1750, it is already well advanced. For the record: Newcomen's machine was developed in 1709. In Belgium, the importance of the mining industry in the 18th century is totally forgotten, as in the 19th century as the importance of the non-ferrous industry ( Zinc, with the very important site of Moresnet). Finally, the authors obviously do not know that in the 17th and 18th centuries, the greatest industrial power in continental Europe was the German-speaking technical space (Holy Roman Empire) and Austria-Hungary), that the main producer of steel and copper was Sweden, and that the industrial revolution began in the United Kingdom, with technicians from these countries. In other words, the "Industrial Revolution" ie, the primacy of the secondary sector over other economic sectors, was based on strong industrialization, around mines, and metallurgy during the 1680-1750s, in UK, Belgium (not in France), and, the textile industry, which went in through a phase of "proto-industrialization" very different from the "pre-industrialization" phase experienced by heavy industry.
Another important point: the quote from K. Marx suggests that there is an automatic association between liberalism and industrialization. It's wrong. in Austria-Hungary, for example, the industry grew upon an authoritarian regime. To summarize: this part needs to be reviewed, because it is scientifically too weak. 

Interest to the readers : who is the target audience? If the target is European or American (North and South) researchers and decision makers specializing in the field, the interest is very low. If the target is researchers from countries currently industrialized or in the process of industrialization (Asia or Africa, Australia), then it should be less descriptive and more analytic and heuristic. Why not highlight a case study: Blégny-les-Mines, for example? Furthermore, in any case, it would have been interesting to show how it was interesting to use the CoMod tool, on this subject.

Author Response

Dear Editor,

Dear Reviewers,

With this cover letter, we will submit the revised Manuscript (information-1021642) entitled ‘The overview of the conservation and renewal of the industrial Belgian heritage as a vector for cultural regeneration’ for publication in the journal ‘Information’. We would like to thank you for the careful and constructive reviews and suggestions. Based on the comments received, we have revised the manuscript and updated its current version.

Editor

It would be appreciated if you rephrase the content to avoid paragraph copy.

The similarity report is attached. Please rephrase the highlighted parts (especially in the abstract, analysis, and conclusion part) in your own words. You can keep the equations and basic concepts as it is.

Reply to Editor’s comments:

Dear Editor,

Thank you for your comment and suggestion, which are well considered at the updated version of our submitted manuscript. An exhaustive revision of each sentence of the manuscript was performed by rephrasing accordingly its context following the guide of your report proposed aiming to reduce the similarity index. The proposed changes are presented to the updated version of the manuscript. All the modifications following your comments and suggestions are included at the submitted version with ‘green’ highlights.

Reviewer 4:

Originality:

Very low. This article could have been written twenty years ago. It evokes the current debates, but in a way too superficial. We would like to know how the experience acquired in the management of industrial heritage has changed the concept and its understanding on the one hand, and the actors involved on the other. From this point of view, the actors who took charge of the enhancement of the industrial heritage are absent from the analysis: public institutions, associations, architects, town planners, etc. More exactly, they are indicated, but without analysis of their role, and how it has evolved over the decades of experience mentioned.

Furthermore, the passage of successive phases: industrialization/deindustrialization / cultural and tourist heritage of industrial sites seems to go without saying. It has not happened. It has been a very complicated way, humanely, economically, institutionally. Since the link between industrial heritage and cultural regeneration, we would like to know what happened to the Pays de Liège, for example, which went through a huge depression due to deindustrialization. 

Significance of Content:

Good. Interesting. But there is a lack of references on this topic from other European countries, mainly Italy, Spain, Portugal (which have been also industrialized and are more touristic than Belgium), and from Germany, or more recently, Czech Republic. And : "Cultural regeneration" probably does not mean the same thing in the Pays de Liège and in Flanders ... Belgium is culturally marked by its historical division between Flemish and Walloons, and as the article suggests, but without going into depth, this division refers to a difference in economic history: in the 19th century, Wallonia had the primacy because of its very strong heavy industry ...

Quality of presentation:

Excellent

Scientific Soundness :

Low. The main flaw here is an inaccurate chronology of the industrialization phase, and an overly narrow reading of the pre-industrial phase. There is here a lack of historical reference on the question. To go quickly: the Industrial Revolution concerns the United Kingdom, and not just England (Britain), and in 1750, it is already well advanced. For the record: Newcomen's machine was developed in 1709. In Belgium, the importance of the mining industry in the 18th century is totally forgotten, as in the 19th century as the importance of the non-ferrous industry (Zinc, with the very important site of Moresnet). Finally, the authors obviously do not know that in the 17th and 18th centuries, the greatest industrial power in continental Europe was the German-speaking technical space (Holy Roman Empire) and Austria-Hungary), that the main producer of steel and copper was Sweden, and that the industrial revolution began in the United Kingdom, with technicians from these countries. In other words, the "Industrial Revolution" ie, the primacy of the secondary sector over other economic sectors, was based on strong industrialization, around mines, and metallurgy during the 1680-1750s, in UK, Belgium (not in France), and, the textile industry, which went in through a phase of "proto-industrialization" very different from the "pre-industrialization" phase experienced by heavy industry.

Another important point: the quote from K. Marx suggests that there is an automatic association between liberalism and industrialization. It's wrong. in Austria-Hungary, for example, the industry grew upon an authoritarian regime. To summarize: this part needs to be reviewed because it is scientifically too weak. 

Interest to the readers: who is the target audience? If the target is European or American (North and South) researchers and decision-makers specializing in the field, the interest is very low. If the target is researchers from countries currently industrialized or in the process of industrialization (Asia or Africa, Australia), then it should be less descriptive and more analytic and heuristic. Why not highlight a case study: Blégny-les-Mines, for example? Furthermore, in any case, it would have been interesting to show how it was interesting to use the CoMod tool, on this subject.

Answer:

Dear Reviewer,

First of all, we would like to thank you for your extensive review and your suggestions. We have considered carefully all your comments and the relevant modifications are included at the updated version of the manuscript submitted. All the modifications following your comments and suggestions are included at the submitted version with ‘blue’ highlights.

In particular:

Originality:

  1. Very low. This article could have been written twenty years ago. It evokes the current debates, but in a way too superficial. We would like to know how the experience acquired in the management of industrial heritage has changed the concept and its understanding on the one hand, and the actors involved on the other. From this point of view, the actors who took charge of the enhancement of the industrial heritage are absent from the analysis: public institutions, associations, architects, town planner, etc. More exactly, they are indicated, but without analysis of their role, and how it has evolved over the decades of experience mentioned.

Reply: Thank you for your comment. This manuscript aims to fill in the overview of Belgium's industrial heritage from the general level, in order to provide a theoretical and case basis for further research. In this paper, we deeply analyzed the development path of Belgium's industrial heritage, and carry out a detailed and in-depth analysis and discussion. In fact, it could be said that this type of research and manuscript could have been written years ago, nonetheless, despite the fact that Belgium is one of the most important European countries with important historical industrial heritage, there are still ‘open issues and debates’ due to the lack of existing inventories and undoubtably its management and conservation, in which few things have been developed. Precisely, the management experience of industrial heritage keeps changing with the concept of industrial heritage and social development. In chapter 2.1, 2.2, 2.4, they show this experience is a constant change, a constant effect. At the end of chapter 3.2, we add an influence and evolution for industrial heritage protection from the PIWB as an industrial heritage protection joiner.

  1. Furthermore, the passage of successive phases: industrialization / deindustrialisation / cultural and tourist heritage of industrial sites seems to go without saying. It has not happened. It has been a very complicated way, humanely, economically, institutionally. Since the link between industrial heritage and cultural regeneration, we would like to know what happened to the Pays de Liège, for example, which went through a huge depression due to deindustrialisation. 

Reply: Thank you for your comment. We completely agree with you that the passage from the different phases of industrialization and deindustrialization is a long process and a complicated process, however, this study is out of the scope of the paper.

The main idea of the paper is to provide historical landmarks of the derelict areas of former industrialized regions of Belgium and to highlight the importance of its conversation and redevelopment as an opportunity for urban renewal. Belgium has a long history of more than 200 years of industrialization, which ‘shaped’ its identity, for instance the arrival of Cockerill family from England (Leeds), which inaugurated the manufacturing machinery for the spinning industry and later the steam engines, and then the development of the iron and steel industry. Another important landmark of this period has been the creation of the railway network in 1834 and the Europe’s first steam-powered railway line between Mechelen and Brussels in 1835 and the case of A. Sauveur, who was distinguished as a ferrous metallurgist. In summary, what should be noticed is that Belgium was the first European country to adopt industrial production (thanks to J. Cockerill), with steel production to become the most important sector and ranking Belgium as one of the highest world producers or iron and steel.

In particular, for the Province (or as you mentioned the ‘Pays’) of Liège[1], in the 19th century, Europe entered the new era of mechanization and the city of Liège became a very important and remarkable industrial center of Belgium linked to mining and the metallurgical activities. In detail, in 1837, we observed the creation of the Vieille-Montagne company leading one of the most important steel-making companies and since then its growth was rapid. Many constructions were developed during this period, for instance, in 1849, the first stone of the Provincial Palace was laid, a new Pont des Arches bridge was inaugurated in 1860 as a protection against the flood risks, while at the same time, a large road networking was developed through the former urban tissue. Furthermore, the île du Commerce isle and its surroundings were transformed: a harbour was built before the construction of the vast Esplanade des Terrasses, Parc d'Avroy and Boulevard Piercot with its monumental music academy (called the Conservatoire), completed in 1886, where Liège experienced its first social riots and becomes the home to ideological conflicts between liberals and Catholics.

During the 20th century, we notice the creation of the autonomous port in 1938 and the International Exhibition on Water (1939). After the end of the Second World war, we observe the recession of the industrial activity in the region and the close of coal mines. Over a twenty-year period, between 1960 and 1980, Liège lost two thirds of its jobs in traditional industry. The service economy progressively replaced it, whilst the steel-making and metallurgical industries, following the merger of several companies, such as Cockerill, Espérance-Longdoz and Arcelor, headed towards the closure in 2009 of its furnaces. During this same period, the city underwent the modernisation of its buildings, the fever for rejuvenation and the penetration of major roads into the heart of the city. Many apartment blocks rose up high into the Liège skyline, especially along the Meuse River, such as the buildings in the Droixhe plain, for example. The years between 1975 and 1990 witnessed a period of downturn. The financial difficulties of the city authorities and the economic crisis in the industrial heartland of the Liège region obliged the city to function on stand-by.

The economic, and therefore the social, redeployment of the city can be illustrated in four main domains: the transformation of metal, the introduction of new technologies, the multi-modal activities and the sector of services and infrastructures. At the turn of 21st century, renovations are coming to the city with the inauguration of the attractive infrastructures of Belle-Île and Médiacité, the radiant renovations of the gems of local architecture – such as the Grand Curtius, the Museum of Walloon Life and the Treasure House of the Cathedral – as well as the new Liège Guillemins railway station offer a very promising outlook for the future. A prime point of convergence of economic flows and major events of civilization on which it has often been able to leave the mark of its specific spirit, at present Liège is seeking its true calling.

Significance of Content:

  1. Interesting. But there is a lack of references on this topic from other European countries, mainly Italy, Spain, Portugal (which have been also industrialized and are more touristic than Belgium), and from Germany, or more recently, Czech Republic. And : "Cultural regeneration" probably does not mean the same thing in the Pays de Liège and in Flanders ... Belgium is culturally marked by its historical division between Flemish and Walloons, and as the article suggests, but without going into depth, this division refers to a difference in economic history: in the 19th century, Wallonia had the primacy because of its very strong heavy industry ...

Reply: Thank you for your comment. In this paper, for the birth, development and structure of Belgium's industry heritage, the influence of neighboring industrialized countries is fully considered. And we compared their differences horizontally and vertically.

Industrial tourism is one of the paths of industrial heritage protection. We do not focus on industrial tourism especially. However, your proposal is acceptable and reasonable. We added the relevant industrial touristic countries' data in industrial tourism at the end of chapter 4, then make a comparison and analysis between these countries. The result shows that compared with neighboring countries, Belgium has the highest density of industrial heritage tourism cases. This result coincides with the motivation and necessity of this study.

Regarding the problematic of the ‘cultural regeneration’, the authors focus on its general context and the importance of renewing the former industrialized regions (such as Liège or the areas of Flanders as you mentioned, too) for diverse reasons, such as the rehabilitation and reuse of these areas but also the increase of their attractiveness as an integral part of the Belgian history. We confess that possibly we can observe differences in the two parts of the country (Walloon and Flanders) and undoubtably on the way they conserve and regenerate their heritage, however, we all agree on the fact that both areas witnessed a period of intense and quite similar industrialization activities, which led to derelict, contaminated and unused areas. Indeed, Wallonia was among the first regions in Northern Europe to industrialize during the 19th century but heavy industrialized activities collapsed meanwhile, as it is also explained in the manuscript, focusing on glass making and coal mining, but Flanders boomed during the postwar era attracting much foreign investments.

Despite their differences in terms of cultural or social approaches, the industrialization was developed in parallel and a strong emphasis on its spatial dimension is crucially important to reveal the opportunities for further and future development. The particularity of Belgium with its division in two parts is (and should not be) not a constraint to develop holistic strategies, tools, policies and approaches for the management of the industrial ‘richness’ of the country and its regeneration, as a whole aiming to promote the cultural regeneration and its future exploitation.

Scientific Soundness :

  1. The main flaw here is an inaccurate chronology of the industrialization phase, and an overly narrow reading of the pre-industrial phase. There is here a lack of historical reference on the question. To go quickly: the Industrial Revolution concerns the United Kingdom, and not just England (Britain), and in 1750, it is already well advanced. For the record: Newcomen's machine was developed in 1709. In Belgium, the importance of the mining industry in the 18th century is totally forgotten, as in the 19th century as the importance of the non-ferrous industry (Zinc, with the very important site of Moresnet). Finally, the authors obviously do not know that in the 17th and 18th centuries, the greatest industrial power in continental Europe was the German-speaking technical space (Holy Roman Empire) and Austria-Hungary), that the main producer of steel and copper was Sweden, and that the industrial revolution began in the United Kingdom, with technicians from these countries. In other words, the "Industrial Revolution" ie, the primacy of the secondary sector over other economic sectors, was based on strong industrialization, around mines, and metallurgy during the 1680-1750s, in UK, Belgium (not in France), and, the textile industry, which went in through a phase of "proto-industrialization" very different from the "pre-industrialization" phase experienced by heavy industry.

Reply: Thank you for your comment, which is considered accordingly. The authors provided more information at the revised version of the manuscript regarding the phase of the pre-industrialization period (sub-section 3.3.1).

  1. Another important point: the quote from K. Marx suggests that there is an automatic association between liberalism and industrialization. It's wrong. in Austria-Hungary, for example, the industry grew upon an authoritarian regime. To summarize: this part needs to be reviewed, because it is scientifically too weak. 

Reply: Thank you for your comment. We performed a more detailed research to reply to your comment and we totally agree on this point that Marx’s quote refers to the relation of liberalism and the industrialization and Belgium has no trace of this movement. Even in the coal and mental mining workers (of both sexes and ages), in ‘perfect freedom’ and at any period and length of time. Belgium had in 1863, compared to 1850, nearly doubled both the amount and the value of its exports of coal, iron, etc. (information retrieved by Marx’s book ‘The Process of Continental Liberalism’). Therefore, we considered that it is better to remove this sentence from the revised version of our manuscript.

  1. Interest to the readers : who is the target audience? If the target is European or American (North and South) researchers and decision makers specializing in the field, the interest is very low. If the target is researchers from countries currently industrialized or in the process of industrialization (Asia or Africa, Australia), then it should be less descriptive and more analytic and heuristic. Why not highlight a case study: Blégny-les-Mines, for example? Furthermore, in any case, it would have been interesting to show how it was interesting to use the CoMod tool, on this subject.

Reply: Thank you for this question, indeed, in this literature review paper, we don’t use an advanced research method. The purpose of this paper is a literature review of Belgian industrial heritage. In this paper, we mentioned there is no similar research in this field. We fill the gap firstly. Then we show the advantages and disadvantages of the protection work in Belgium. This paper could be used for any others who want to do any further research within a Belgian industrial heritage sample, not only researchers but also decision-makers. Meanwhile, we add a suggestion in the conclusion for decision-makers, to make it more analytic and heuristic.

This paper explores the development potential of Belgium's industrial heritage, classifies and sorts the industrial tourism resources, clarifies the potential resource advantages, and determines the priority investment direction. For the currently industrialized or in the process of industrialization countries, there are less of similar overviews also, for instance, in China, there is not an overview of the industrial heritage protection from the scope of the whole country. This paper could be an example for them. For Bleny-les-Mines, thank you for this comment, we expand the part of Bleny-les-Mines. The importance of its existence to Belgium's industrial heritage system is emphasized/highlight once again.

The CoMod is a multi-filed project. It is not only including mathematical modeling but also including other fields in urban planning, such as density. We added some suggestions dealing with the topic of ‘density’, which is the ‘core’ research interest of the project and it is related to the industrial heritage in Belgium based on your previous comment.

We appreciate your comments and we hope answering to all your comments accordingly.

Thank you for reviewing our manuscript.

[1] Information retrieved by : 'History of Liège', Micheline Josse. Second edition expanded and reviewed by Claudine Schloss @ City of Liège - 2009. Modern photographs: Marc Verpoorten

Round 2

Reviewer 2 Report

After an extensive reading of the revised manuscript, I found that the Authors made an extensive major revision as it was suggested. They fully accepted all suggestions and made all proper revisions.

I am fully covered by the answers in my comments and also the revised submitted manuscript is informative and well documented.

I have no further comments to submit.

I would like to see in the near future a second article with an economic analysis of the impact of cultural regeneration on the local economies, e.g. a deeper analysis and correlation of a city’s income/growth with the visitors (tourists).

Author Response

Thank you for your comments.